# Muscle contraction is required to maintain the pool of muscle progenitors via YAP and NOTCH during fetal myogenesis

Joana Esteves de Lima[1,2,3,4], Marie-Ange Bonnin[1,2,3,4], Carmen Birchmeier[5], Delphine Duprez[1,2,3,4]*

[1]CNRS UMR 7622, F-75005 Paris, France; [2]Sorbonne Universités, UPMC Univ Paris 06, Paris, France; [3]IBPS-Developmental Biology Laboratory, Paris, France; [4]Inserm U1156, F-75005, Paris, France; [5]Developmental Biology, Max-Delbrück-Center for Molecular Medicine, Berlin, Germany

**Abstract** The importance of mechanical activity in the regulation of muscle progenitors during chick development has not been investigated. We show that immobilization decreases NOTCH activity and mimics a NOTCH loss-of-function phenotype, a reduction in the number of muscle progenitors and increased differentiation. Ligand-induced NOTCH activation prevents the reduction of muscle progenitors and the increase of differentiation upon immobilization. Inhibition of NOTCH ligand activity in muscle fibers suffices to reduce the progenitor pool. Furthermore, immobilization reduces the activity of the transcriptional co-activator YAP and the expression of the NOTCH ligand *JAG2* in muscle fibers. YAP forced-activity in muscle fibers prevents the decrease of *JAG2* expression and the number of PAX7+ cells in immobilization conditions. Our results identify a novel mechanism acting downstream of muscle contraction, where YAP activates *JAG2* expression in muscle fibers, which in turn regulates the pool of fetal muscle progenitors via NOTCH in a non-cell-autonomous manner.

*For correspondence: delphine. duprez@upmc.fr

Competing interests: The authors declare that no competing interests exist.

## Introduction

Skeletal muscle development, growth and regeneration rely on muscle stem cells. A major goal of muscle research is to understand the signals that regulate the ability of stem cells to self-renew or differentiate.

Skeletal muscle formation involves successive and overlapping phases of embryonic, fetal, perinatal and adult myogenesis. The paired homeobox transcription factors, PAX3 and PAX7, define the pool of muscle stem cells during developmental, postnatal and regenerative myogenesis (*Gros et al., 2005*; *Kassar-Duchossoy, 2005*; *Relaix et al., 2005*). Fetal myogenesis depends on PAX7-expressing muscle progenitors and is associated with muscle growth (*Hutcheson et al., 2009*; *Kassar-Duchossoy, 2005*; *Relaix et al., 2005*). Muscle progenitors undergo myogenic differentiation program with the activation of the bHLH Myogenic Regulatory Factors (MRFs), *MYF5, MRF4, MYOD, MYOG* (*Tajbakhsh, 2009*). By the end of fetal myogenesis, PAX7+ cells adopt a satellite cell position under the basal lamina of muscle fibers (*Biressi et al., 2007*; *Bröhl et al., 2012*). During development, mechanical forces generated by muscle contraction are essential for the correct establishment of the musculoskeletal system. Although the influence of the mechanical forces for cartilage, joint, and bone development has been previously addressed (*Nowlan et al., 2010*; *Rolfe et al., 2014*; *Shwartz et al., 2013*), the consequence of muscle-induced mechanical load for the development of muscle itself is largely unknown.

**eLife digest** Skeletal muscle is attached to the skeleton and allows the body to move. Making a new muscle, or repairing an existing one, relies on stem cells that are present inside muscles. A major goal of skeletal muscle research is to understand the signals that regulate the abilities of muscle stem cells to divide and give rise to more stem cells or to become muscle cells. Molecular signals are known to regulate the numbers of stem cells in the muscle. Skeletal muscles become larger if they are exercised, but it is not clear if mechanical forces generated by muscle contractions directly affect the number of muscle stem cells.

The NOTCH signaling pathway contributes to maintaining the population of stem cells in muscles by forcing the stem cells to divide and preventing them from becoming muscle cells. Here, Esteves de Lima et al. investigated whether muscle contraction regulates NOTCH signaling during muscle formation in chick fetuses.

The experiments show that muscle contraction stimulates the activity of a protein called YAP in muscle cells, which in turn, activates a gene in the NOTCH signaling pathway known as *JAG2*. This increases NOTCH signaling activity in the neighboring stem cells and maintains the number of stem cells in the muscle. The next step following this work will be to establish if this mechanism also operates during muscle formation and regeneration in other animals such as mice and zebrafish.

The NOTCH signaling pathway is a central regulator of skeletal muscle stem cells during embryonic, fetal and adult myogenesis [reviewed in *Mourikis and Tajbakhsh (2014)*]. Activation of the NOTCH signaling pathway requires direct cell-cell contact between a signal-sending cell that expresses the NOTCH ligand and a signal-receiving cell that expresses the NOTCH receptor. Upon ligand activation, the intracellular domain of the NOTCH receptor is cleaved, translocates into the nucleus, associates with the transcription factor RBPJ and activates the transcription of the bHLH transcriptional repressor genes, *HES* and *HEY* [reviewed in *Andersson et al. (2011)*]. In adult myogenesis, NOTCH is involved in satellite cell activation, proliferation and quiescence [reviewed in *Mourikis and Tajbakhsh (2014)*] and the absence of NOTCH signaling in muscle stem cells results in satellite cell depletion due to premature differentiation (*Bjornson et al., 2012*). In addition, during development, NOTCH has been described to activate embryonic myogenesis in somites (*Rios et al., 2011*). During developmental myogenesis, active NOTCH signaling is associated with proliferating muscle progenitors, while NOTCH ligands are expressed in differentiated muscle cells (*Delfini et al., 2000*; *Vasyutina et al., 2007*). NOTCH loss-of-function experiments in mice induce a loss of the muscle progenitor pool due to premature muscle differentiation (*Bröhl et al., 2012*; *Czajkowski et al., 2014*; *Schuster-Gossler et al., 2007*; *Vasyutina et al., 2007*), whereas NOTCH activation represses muscle differentiation in chick and mouse embryos (*Delfini et al., 2000*; *Hirsinger et al., 2001*; *Mourikis et al., 2012*). While studies have identified NOTCH target genes in fetal muscle progenitors (*Bröhl et al., 2012*; *Mourikis et al., 2012*), upstream regulators of NOTCH signaling during developmental myogenesis have not attracted attention.

Similarly to NOTCH, the co-transcriptional activator YAP (Yes-Associated Protein) promotes satellite cell proliferation and inhibits muscle differentiation in culture (*Judson et al., 2012*; *Watt et al., 2010*). In addition to being a nuclear effector of the Hippo pathway, YAP has been identified as a sensor of mechanical activity and mediates cellular and transcriptional responses downstream of mechanical forces (*Aragona et al., 2013*; *Dupont et al., 2011*; *Porazinski et al., 2015*). In addition to other transcription factors, YAP binds to TEAD DNA binding proteins (*Varelas, 2014*). YAP and TEAD1 have been shown to occupy 80% of the same promoters in human mammary epithelial cells (*Zhao et al., 2008*), indicating that YAP/TEAD interaction could constitute the major molecular mechanism of YAP-mediated regulation of gene transcription. The TEAD transcription factors recognize and bind to MCAT elements (CATTCC), which are enriched in regulatory regions of muscle-related genes [reviewed in *Wackerhage et al. (2014)*]. In addition to being involved in muscle stem cell proliferation (*Judson et al., 2012*; *Watt et al., 2010*), YAP has been recently shown to be a critical regulator of skeletal muscle fiber size in adult mice (*Goodman et al., 2015*; *Watt et al., 2015*).

A link between mechanical forces (provided by muscle contraction) and signaling pathways that regulate fetal myogenesis has not been established. In this study, we show the importance of mechanical forces in the regulation of the number of fetal muscle progenitors. We show that immobilization induces a NOTCH loss-of-function phenotype in muscles. We further provide evidence that, downstream of mechanical forces, YAP positively regulates the expression of the NOTCH ligand *JAG2* in fibers, which maintains the pool of muscle progenitors by activating NOTCH signaling.

## Results

### Inhibition of muscle contraction reduces the number of progenitors and increases their differentiation in fetal muscles

To study the effect of mechanical signals on muscle progenitors, we set up an unloading model during chick fetal myogenesis (*Figure 1A*). We used decamethonium bromide (DMB), which blocks muscle contraction and induces rigid paralysis that leads to immobilization (*Nowlan et al., 2010*). Two days after the inhibition of muscle contraction, we observed a reduction in the overall muscle size (*Figure 1D,H*), which is consistent with previous reports (*Crow and Stockdale, 1986*; *Hall and Herring, 1990*). In addition, we observed a decrease of 58.07% (±17.66) in the number of PAX7+ muscle progenitors in paralyzed compared to control muscles (*Figure 1B,D-F,H-J*). Consistently, the relative expression levels of *PAX7* and *MYF5* genes were significantly decreased in DMB-treated limbs compared to control limbs, as early as 12 hr after DMB application and with a more prominent effect at 48 hr (*Figure 1C*). In addition to the reduction of the muscle progenitor pool, we also observed an increase of myogenic differentiation assessed by an increase of *MYOD* and *MYOG* expression using RT-q-PCR or in situ hybridization in immobilized limbs compared to control limbs (*Figure 1C,G,K*). The number of *MYOD*-expressing cells was also increased in paralyzed muscles compared to control muscles (*Figure 1M*). In addition, muscle fibers were larger in limb muscles of DMB-treated fetuses compared to control muscles (*Figure 1E,I*). The large muscle fibers were associated with several nuclei in paralyzed muscles, while control muscle fibers displayed only one nucleus on transverse sections (*Figure 1L*). Injection of pancuronium bromide (PB), a drug that exerts a flaccid skeletal muscle paralysis (*Nowlan et al., 2010*) led to a similar but less pronounced effect, i.e. a 28.92% (±7.27) reduction in the number of PAX7+ cells and a concomitant increase of muscle differentiation (*Figure 1—figure supplement 1*). DMB or PB treatments of chick fetal myoblast cultures did neither affect muscle progenitors nor their differentiation and did not change the expression levels of *PAX7*, *MYF5*, *MYOD* and *MYOG* (*Figure 1—figure supplement 2*), indicating that DMB and PB did not have any off-target effect on myogenic cells. We conclude that the inhibition of muscle contraction leading to rigid or flaccid paralysis reduces the pool of fetal muscle progenitors and increases their propensity to differentiate.

### NOTCH activity is decreased in muscles of immobilized fetuses

The concomitant decrease of the muscle progenitor pool and increase of muscle differentiation following muscle paralysis was reminiscent of a NOTCH loss-of-function phenotype. In the murine system, loss of NOTCH signaling results in a reduction of the progenitor pool due to a precocious shift toward differentiation (*Bröhl et al., 2012*; *Vasyutina et al., 2007*). To determine whether NOTCH activity was modified upon immobilization, we examined the expression of components of the NOTCH signaling pathway in paralyzed muscles. During fetal myogenesis, the NOTCH ligand *JAG2* was expressed in MF20+ differentiated muscle cells (*Delfini et al. 2000*), while *HES5* a recognized transcriptional readout of NOTCH activity (*Andersson et al., 2011*) was excluded from differentiated muscle fibers (*Figure 2A,D*). The NOTCH ligand *DLL1* is expressed during chick and mouse embryonic myogenesis (*Vasyutina et al. 2007*, *Delfini et al., 2000*) but is not detected by in situ hybridization in chick limb fetal muscles (*Delfini et al., 2000*). *JAG2* and *HES5* were also expressed in blood vessels (*Figure 2B,E*, arrowheads). In immobilized fetuses, the expression of the *JAG2* and *HES5* was decreased in paralyzed muscles (*Figure 2C,F,I*) compared to control muscles (*Figure 2B,E,H*). We believe that the downregulation of *JAG2* and *HES5* expression in paralyzed muscles reflected a muscle-specific loss of gene expression, since *JAG2* and *HES5* expression was not affected in blood vessels (*Figure 2B,C,E,F,H,I*, arrowheads). Moreover, blood vessels are surrounding fetal muscles in

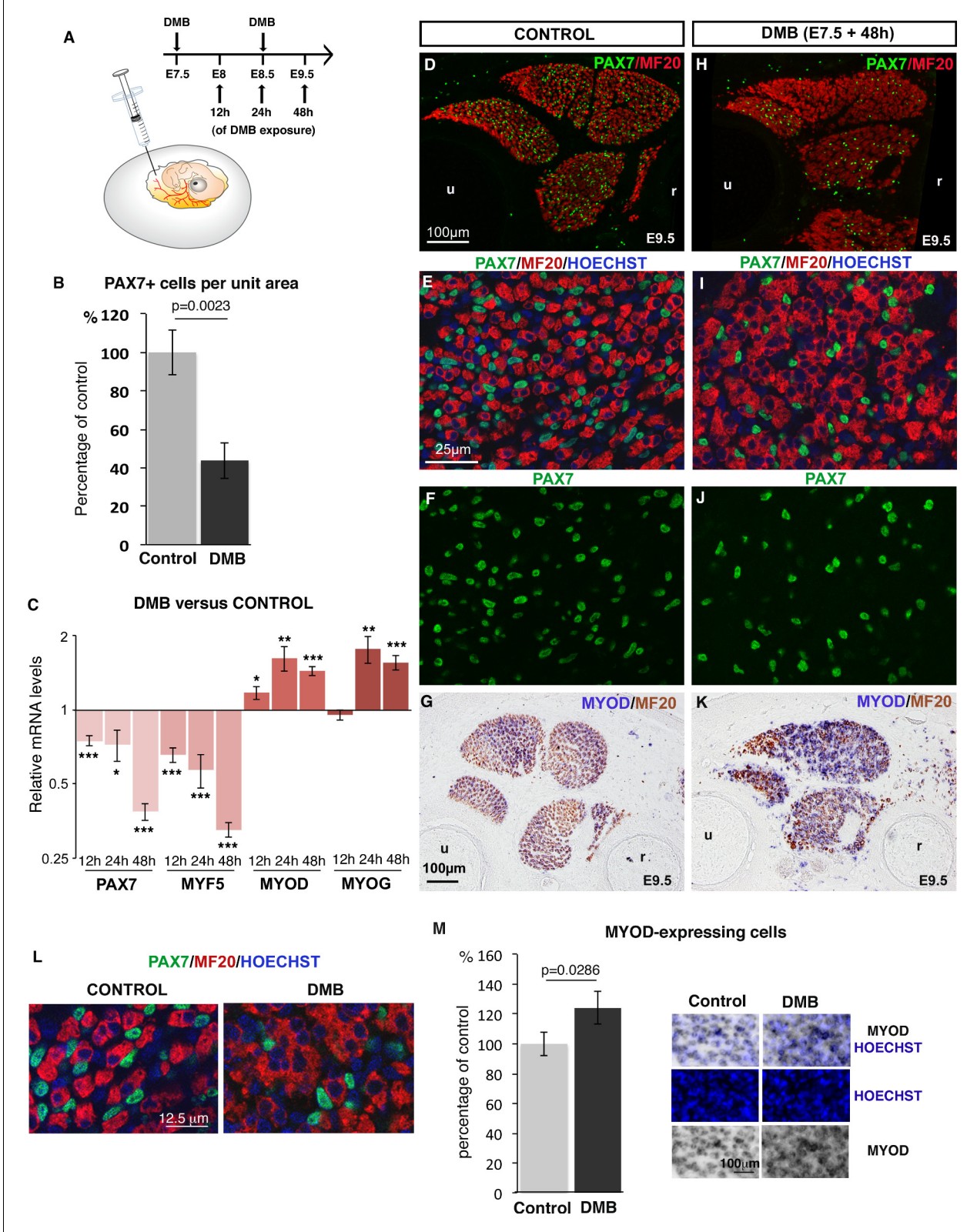

**Figure 1.** Inhibition of muscle contraction decreases the number of limb fetal muscle progenitors. (**A**) Chick fetuses were treated with DMB at E7.5 and E8.5, in order to block muscle contraction. (**B**) Number of PAX7+ cells in paralyzed and control muscles. PAX7+ cell number was counted per unit area in dorsal and ventral muscles of three DMB limbs and three control limbs. Results are shown as percentage of control. Error bars indicate standard deviations. The p-value was obtained using the Mann-Withney test. (**C**) RT-q-PCR analyses of muscle gene expression levels in limbs 12 hr, 24 hr and

*Figure 1 continued on next page*

*Figure 1 continued*

48 hr after DMB treatment compared to control limbs. For each gene, the mRNA levels of control limbs were normalized to 1. Graph shows means ± standard errors of the mean of 11 limbs. The p-values were calculated using the Wilcoxon test. Asterisks indicate the p-values *p<0.05, **p<0.01, ***p<0.001. (D–K) Control limbs (D–G) (N = 5) and limbs from DMB-treated embryos (H–K) (N = 5) were transversely sectioned at the level of the zeugopod and analyzed for muscle progenitors and differentiated cells by immunohistochemistry using the PAX7 and MF20 antibodies, respectively (D–F, H–J), or for *MYOD* expression by in situ hybridization followed by immunohistochemistry with MF20 antibody (G,K) (N = 4). Nuclei were labeled with Hoechst (blue). Limb sections are dorsal to the top and posterior to the left. u, ulna, r, radius. (L) High magnifications of muscle fibers to show the grouped nuclei in fibers of paralyzed muscles compared to control muscles. (M) Number of Hoechst+ nuclei of *MYOD*-expressing cells versus all Hoechst+ nuclei. Results are shown as percentage of control. Error bars indicate standard deviations. The p-value was obtained using the Mann-Withney test.

The following figure supplements are available for figure 1:

**Figure supplement 1.** Muscle flaccid paralysis decreases the number of PAX7+ muscle progenitors and increases their differentiation.

**Figure supplement 2.** DMB or PB treatment in primary cultures of chick fetal myoblasts did not change the expression of muscle genes.

normal conditions (*Figure 2J,K*), when muscle splitting is accomplished (*Tozer et al., 2007*). Consistently, the *JAG2* and *HES5* mRNA levels were moderately but significantly downregulated in limbs of immobilized animals (*Figure 2G*). We believe that the unchanged *JAG2* and *HES5* expression in blood vessels obscures the changes in *JAG2/HES5* expression levels in muscles. *HeyL* is another transcriptional readout of NOTCH that responds to NOTCH activation in limb fetal muscle cells in mice (*Mourikis et al., 2012*; *Bröhl et al., 2012*). The relative mRNA levels of *HEYL* gene were also significantly downregulated in limbs of immobilized animals compared to controls (*Figure 2G*). In summary, NOTCH activity was reduced in paralyzed muscles, as assessed by reduced expression of the NOTCH ligand *JAG2* in muscle fibers and of the transcriptional readout of NOTCH activity, *HES5*, in muscles. The downregulation of NOTCH signaling (*Figure 2*) and the concomitant reduction of the muscle progenitor pool and increase of differentiation (*Figure 1*, *Figure 1—figure supplement 1*) observed in unloading conditions led us to conclude that immobilization mimics a NOTCH loss-of-function phenotype in muscles.

## The absence of muscle contraction does not affect muscle progenitor proliferation

As the pool of muscle progenitors was decreased after muscle paralysis (*Figure 1*), we assessed proliferation and apoptosis of muscle progenitors by EdU incorporation and TUNEL analyses in immobilization conditions (*Figure 3*). EdU incorporation into PAX7+ cells was not altered, indicating that the proliferative rate of PAX7+ cells was not changed during immobilization compared to control conditions (*Figure 3A–G*). Despite the reduced number of PAX7+ cells in paralyzed muscles (*Figure 1*), their proliferation capacities were unaffected (*Figure 3A–G*). While apoptotic cells were very rare in control muscles (*Figure 3H,J,L,N*), we observed an increase of apoptotic figures in fetal muscles of immobilized animals, 48 hr after DMB treatment (*Figure 3I,K,M,O*). We found no increase of apoptosis 24 hr after DMB treatment (data not shown). The apoptotic figures were not observed in PAX7+ cells, 48 hr after DMB treatments (*Figure 3H–K*), nor in MF20+ cells or in Desmin+ cells (*Figure 3J–O*). The absence of apoptotic figures in myogenic cells suggests that cells undergoing apoptosis upon immobilization are possibly muscle connective tissue cells. Thus, immobilization did neither affect the proliferative capacity nor apoptosis of PAX7+ cells, which was reminiscent of the unchanged proliferation and apoptosis rate of muscle progenitors in NOTCH loss-of-function in mice (*Vasuytina et al., 2007*).

## Ligand-mediated activation of NOTCH prevented the loss of muscle progenitors in the absence of muscle contraction

The inhibition of muscle contraction mimicked a NOTCH loss-of-function phenotype in fetal muscles (*Figures 1–3*). To determine whether NOTCH signaling acts downstream of mechanical signals to regulate the muscle progenitor pool, we performed NOTCH rescue experiments in immobilization conditions. DLL1-induced NOTCH activation in chick embryos inhibits muscle differentiation

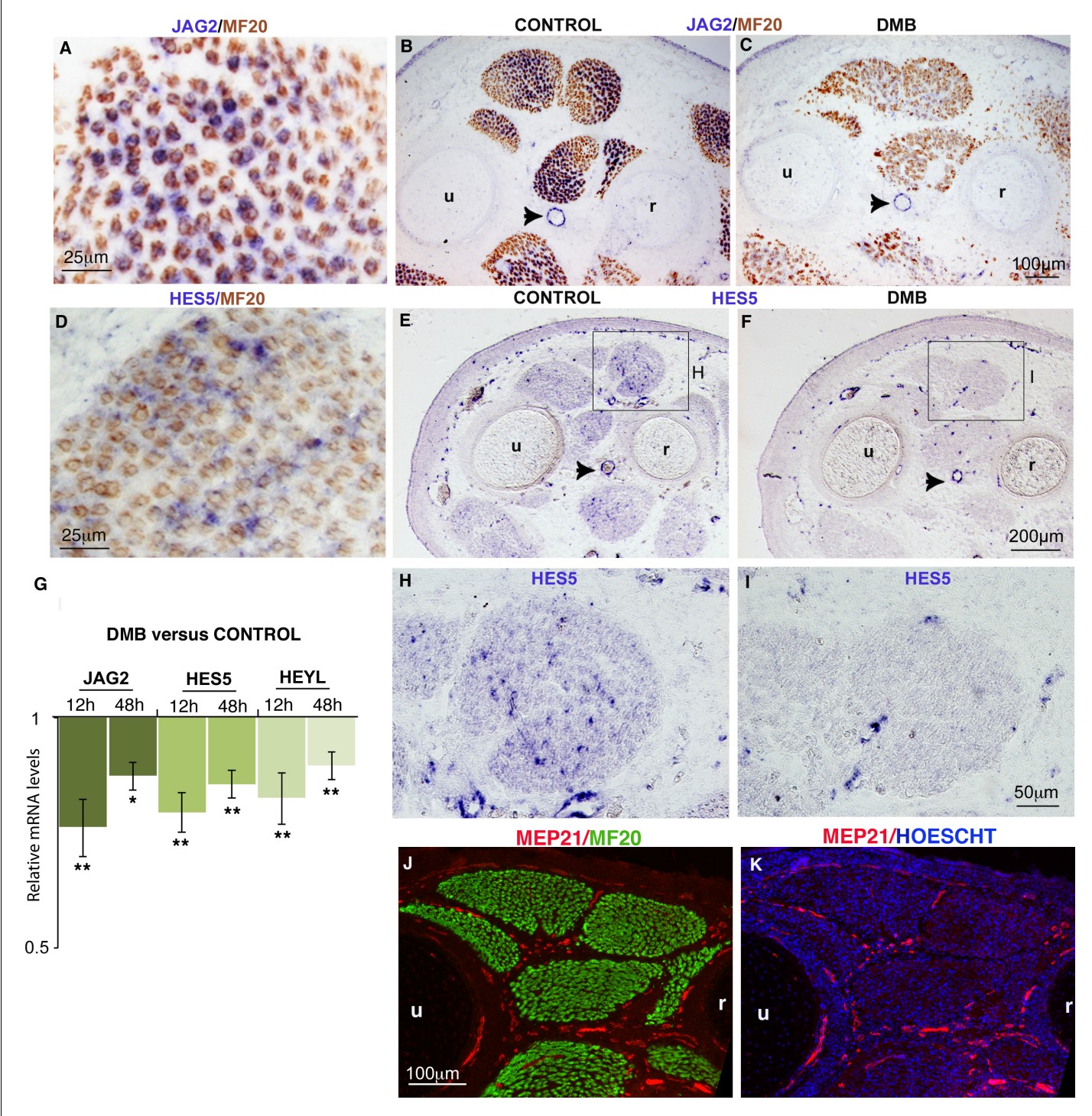

**Figure 2.** Muscle contraction inhibition decreases NOTCH activity in limb muscles. (**A–F, H–I**) In situ hybridization to transverse limb sections at the level of the zeugopod of E9.5 chick fetuses treated (**C,F,I**) or not treated (**A,B,D,E,H**) with DMB, with JAG2 (**A–C**) or HES5 (**D–F,H,I**) probe followed by immunostaining with MF20 antibody (**A–D**) (N = 4). The expression of the NOTCH ligand *JAG2* was lost in muscle fibers of paralyzed muscles (**C**) compared to *JAG2* normal expression (**A,B**). (**B,C**, arrowheads) *JAG2* expression in blood vessels was not affected in DMB limbs. The expression of the transcriptional readout of NOTCH, *HES5* was also decreased in paralyzed muscles (**F,I**) compared to control muscles (**D,E,H**). (**G**) RT-q- PCR analyses of the expression levels of NOTCH signaling components in limbs of 12 hr and 48 hr DMB-treated fetuses. For each gene, the mRNA levels of control limbs were normalized to 1. Graph shows means ± standard errors of the mean of eight limbs. The p-values were calculated using the Wilcoxon test. Asterisks indicate the p-values, *p<0.05, **p<0.01. (**J,K**) Transverse limb sections of E9.5 fetuses were immunostained with MEP21 and MF20 antibodies

*Figure 2 continued on next page*

*Figure 2 continued*

to visualize blood vessels and differentiated muscles, respectively (N = 3). Hoescht was used to visualize nuclei in blue. Limb sections are dorsal to the top and posterior to the left. u, ulna, r, radius.

(*Delfini et al., 2000*; *Hirsinger et al., 2001*). We first performed ligand-dependent NOTCH activation and analyzed the consequences for PAX7+ cells in normal conditions of muscle activity. In *DLL1*-activated NOTCH limbs, the pool of PAX7+ cells was maintained at E9.5, despite inhibition of muscle differentiation (*Figure 4—figure supplement 1*), which is consistent with the maintenance of PAX7+ progenitors in NICD-expressing cells in mouse fetuses (*Mourikis et al., 2012*). Thus, ligand-activated NOTCH maintained the number of chick fetal muscle progenitors over time and impaired their differentiation. We then forced NOTCH activity in limbs of immobilized fetuses and analyzed the consequences for muscle using the contralateral non-grafted limbs as control (*Figure 4*). Under immobilization conditions, the left limbs displayed a reduction in progenitor numbers and increased differentiation (*Figure 4A,C,E,G*). In contrast, in DLL1-expressing regions of (right) limbs, the number of PAX7+ cells was increased by 215.81% (± 29.36) and myosin expression was decreased compared to control (left) limbs (*Figure 4A–I*). Since immobilization leads to a 2.3-fold (58.07%) reduction in the number of PAX7+ cells (*Figure 1B*), we estimated that retroviral DELTA1-induced NOTCH rescued around 93% the number of PAX7+ cells in paralyzed muscles. We conclude that NOTCH activation prevents the decrease in the number of PAX7+ cells by preventing their inappropriate differentiation under immobilization conditions.

## NOTCH ligand activity in differentiated muscle cells is required to maintain the pool of muscle progenitors

NOTCH signaling depends on the direct contact between signal-sending cells that express the NOTCH ligands and signal-receiving cells that express NOTCH receptors and display active NOTCH (*Andersson et al., 2011*). Since we observed a concomitant loss of the expression of the NOTCH ligand *JAG2* in muscle fibers and reduced expression of NOTCH target genes in paralyzed muscles (*Figure 2*), it was unclear which cell type, between differentiated fibers and progenitors, first sensed muscle contraction. To test whether the loss of NOTCH ligand in muscle fibers would suffice to reduce the size of the muscle progenitor pool, we blocked NOTCH ligand function specifically in muscle fibers. For this, we overexpressed a dominant-negative form of DELTA1 (DELTA1/DN) (*Figure 5A*), which prevents NOTCH ligand processing in signal-sending cells and therefore blocked NOTCH activation in signal-receiving cells (*Chitnis, 2006*; *Henrique et al., 1997*). We performed chick somite-electroporation at the forelimb level (*Figure 5B*) using a stable vector that can be integrated into the genome in the presence of a transposase (*Bourgeois et al., 2015*). This vector co-expresses the Tomato reporter gene and DELTA1/DN under the control of the mouse Myosin Light Chain (MLC) promoter that drives expression in chick-differentiated muscle cells and not in muscle progenitors (*Wang et al., 2011*). In electroporated Tomato-expressing muscles, 44.94% (±12.88) of MF20+ cells displayed Tomato expression (*Figure 5D–E*). We observed that the lack of ligand activity in around 45% of differentiated muscle cells significantly decreased the number of PAX7+ cells by 32.42% (±7.23) compared to contralateral limbs (*Figure 5F–K*). This shows that a diminution of NOTCH ligand activity in muscle fibers suffices to induce a decrease in the number of PAX7+ cells. We conclude that NOTCH ligand activity in muscle fibers is required to maintain the pool of fetal muscle progenitors.

## YAP activity is observed in differentiated muscle fibers and is decreased in paralyzed muscles

We next aimed to identify the signal that could sense mechanical forces and regulate *JAG2* expression in muscle fibers. We focused on the transcriptional co-activator YAP that has been shown to sense mechanical signals independently of the Hippo pathway (*Aragona et al., 2013*; *Dupont et al., 2011*). *YAP1* transcripts and YAP protein were expressed ubiquitously in chick limbs (*Figure 6—figure supplement 1A–E*). Since the subcellular localization of YAP protein reflects its transcriptional activity (*Dupont et al., 2011*), we examined nuclear YAP staining in MF20+ and PAX7+ cells.

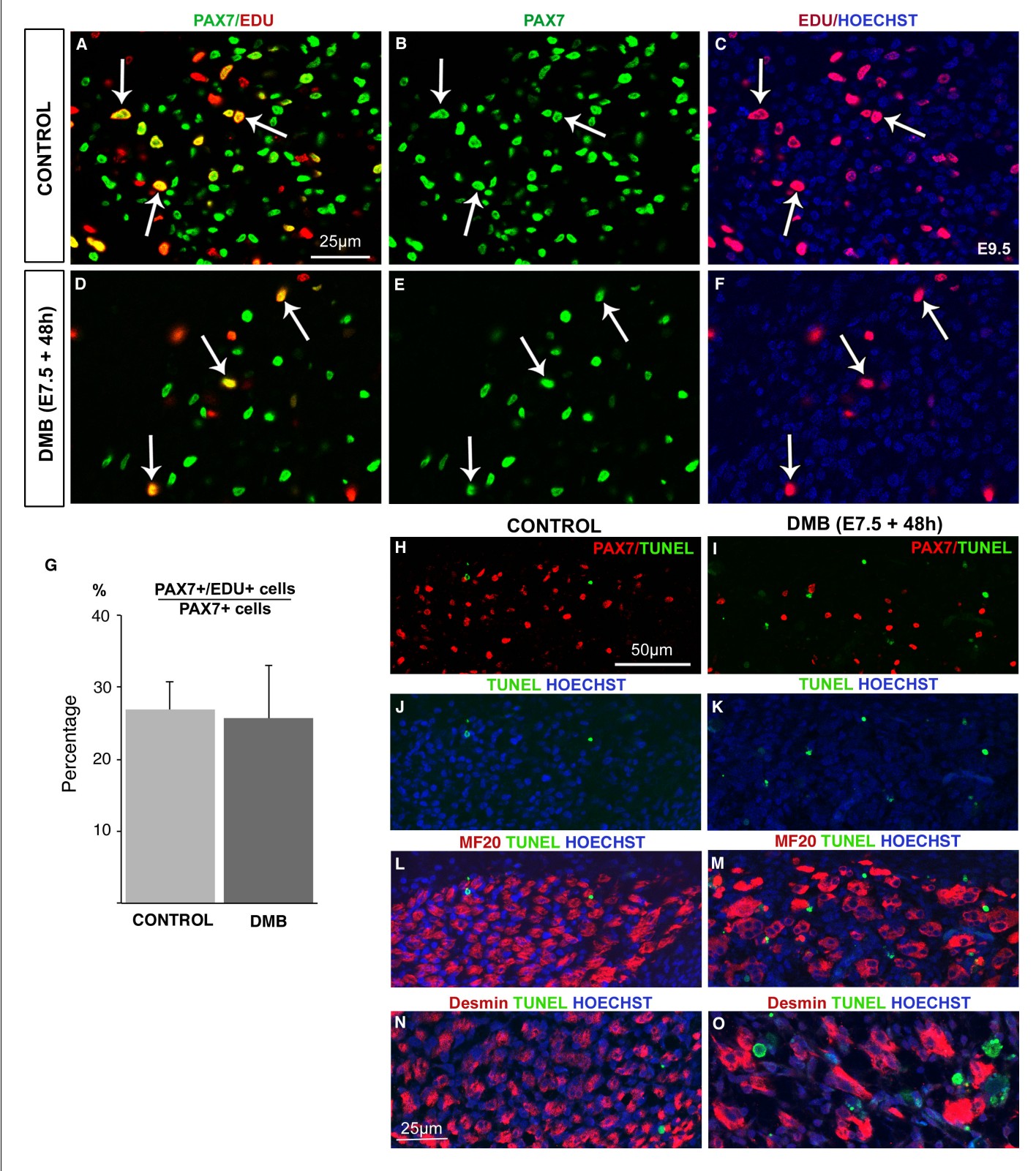

**Figure 3.** The proliferation rate of muscle progenitors is not modified in paralyzed muscles. Limb muscles of control (**A–C**) and DMB-treated (**D–F**) fetuses were analyzed for cell proliferation by EdU incorporation. Control and paralyzed muscles displayed EdU+/PAX7+ cells showing proliferating muscle progenitors (**A–F**, arrows). The percentage of EdU+/PAX7+ cells on PAX7+ cells was analyzed in three muscles of three DMB limbs and three control limbs. (**G**) The percentage of EdU+/PAX7+ cells on PAX7+ cells was similar in both control and paralyzed muscles. Error bars show standard

*Figure 3 continued on next page*

Figure 3 continued

deviations. Control (H,J,L,N) and paralyzed (I,K,M,O) muscles were analyzed for apoptosis. TUNEL staining was rarely visualized in control muscles (H,J, L,N), while paralyzed muscles displayed an increase of apoptotic figures (I,K,M,O), which were not located in PAX7+ cells (I), in Myosin+ cells (M) or in Desmin+ cells (O).

Unexpectedly, given the proliferative role of YAP in organ formation and tumorigenesis (*Zanconato et al. 2015*, *Piccolo et al., 2014*), we found that 89.6% (±3.65) of myonuclei of post-mitotic MF20+ cells displayed nuclear YAP protein (*Figure 6A-C,G,H*; *Figure 6—figure supplement 1F,G*). Moreover, YAP nuclear staining was stronger than YAP cytoplasmic staining in MF20+ cells; the cytoplasmic domains being delineated with MF20 staining (*Figure 6G*, *Figure 6—figure supplement 1F*). In contrast, only a subset of PAX7+ cells showed nuclear YAP staining (*Figure 6—figure supplement 1H*). Myoblast proliferation has been shown to be associated with increased nuclear YAP in cell cultures (*Judson et al., 2012*; *Watt et al., 2010*). However, we did not detect an obvious correlation between nuclear YAP protein and the proliferative state of PAX7+ cells in chick limb fetal muscles, in vivo (*Figure 6—figure supplement 1I*). ANKRD1 and CTGF are two recognized direct target genes of YAP in many cell types (*Lai et al., 2011*; *Zhao et al., 2008*). ANKRD1 is a muscle ankyrin-repeat protein that binds sarcomeric proteins (*Kojic et al., 2011*), and CTGF is a secreted matricellular protein involved in multiple cellular processes (*Malik et al., 2015*). ANKRD1 and CTGF expression was observed at the tips of MF20+ muscle fibers (*Figure 6I-M,Q*). In fetal limbs, ANKRD1 was exclusively expressed in muscle fibers (*Figure 6I,K,L*), whereas CTGF transcripts were observed also in cartilage (*Figure 6M,Q*). YAP staining in myonuclei and the expression of YAP target genes in muscle fibers show that YAP is active in differentiated muscle cells of chick fetal limbs. The regionalized location of the YAP target gene transcripts at muscle tips along fibers suggests that additional proteins are regionalized at muscle tips to regulate ANKRD1 and CTGF transcription. In immobilized fetuses, ANKRD1 and CTGF expression was lost in muscle fibers (*Figure 6I–R*), but CTGF expression was maintained in cartilage (*Figure 6M,P–R*). Consistently, the mRNA levels of ANKDR1 showed a drastic diminution, while those of CTGF displayed a significant but less strong reduction in immobilization conditions (*Figure 6S*). CYR61 is another matricellular protein of the same family as CTGF and is also a YAP target gene (*Lai et al., 2011*). CYR61 mRNA levels were also decreased in limbs 48 hr after DMB treatment (*Figure 6S*). Moreover, the number of YAP+ myonuclei was significantly decreased in muscle fibers of paralyzed muscles compared to control muscles (dropping from 89.6% ± 3.66 to 15.9% ± 4.36) (*Figure 6H*). We did not observe any obvious modification of nuclear YAP staining in PAX7+ cells of DMB-treated versus control animals (*Figure 6—figure supplement 1J,K*), indicating that the nuclear YAP staining in the subpopulation of PAX7+ cells is unrelated to mechanical signals. We conclude that YAP activity is observed in muscle fibers of contracting muscles and that this activity is lost in paralyzed muscles.

## YAP forced-activity in differentiated muscle cells rescues *JAG2* expression and the muscle phenotype in paralyzed muscles

The concomitant loss of YAP activity and *JAG2* expression in muscle fibers of paralyzed muscles prompted us to test whether YAP controlled *JAG2* expression. We overexpressed chick fetal myoblasts in a constitutively active form of mouse Yap that cannot be phosphorylated at Ser112 and is therefore preferentially translocated to the nucleus (*Xin et al., 2013*). Transfection of YapS112A/RCAS into fetal myoblasts increased the expression levels of YAP target genes (*CTGF* and *CYR61*) (*Figure 7—figure supplement 1A*). In addition, we also observed changes in muscle gene expression, previously described in C2C12 cells and satellite cell-derived myoblasts (*Judson et al., 2012*; *Watt et al., 2010*), i.e. concomitant increase of *PAX7* and *MYF5* and reduction of *MYOD* and *MYOG* expression. YapS112A also activated *JAG2* expression and NOTCH target genes (*Figure 7—figure supplement 1A*). *JAG2* expression was also upregulated in differentiated muscle cells after transfection with MLC-Tomato-YapS112A (Tomato+ FACS-sorted cells) (*Figure 7—figure supplement 1B*). Thus, YAP positively regulated *JAG2* expression in cultured fetal muscle cells. In order to assess whether YAP would rescue *JAG2* expression and the muscle phenotype observed in immobilization conditions, in vivo, we electroporated the MLC-Tomato-YapS112A construct in chick limb somites of embryos that were then treated with DMB. We observed that YAP forced-activity in

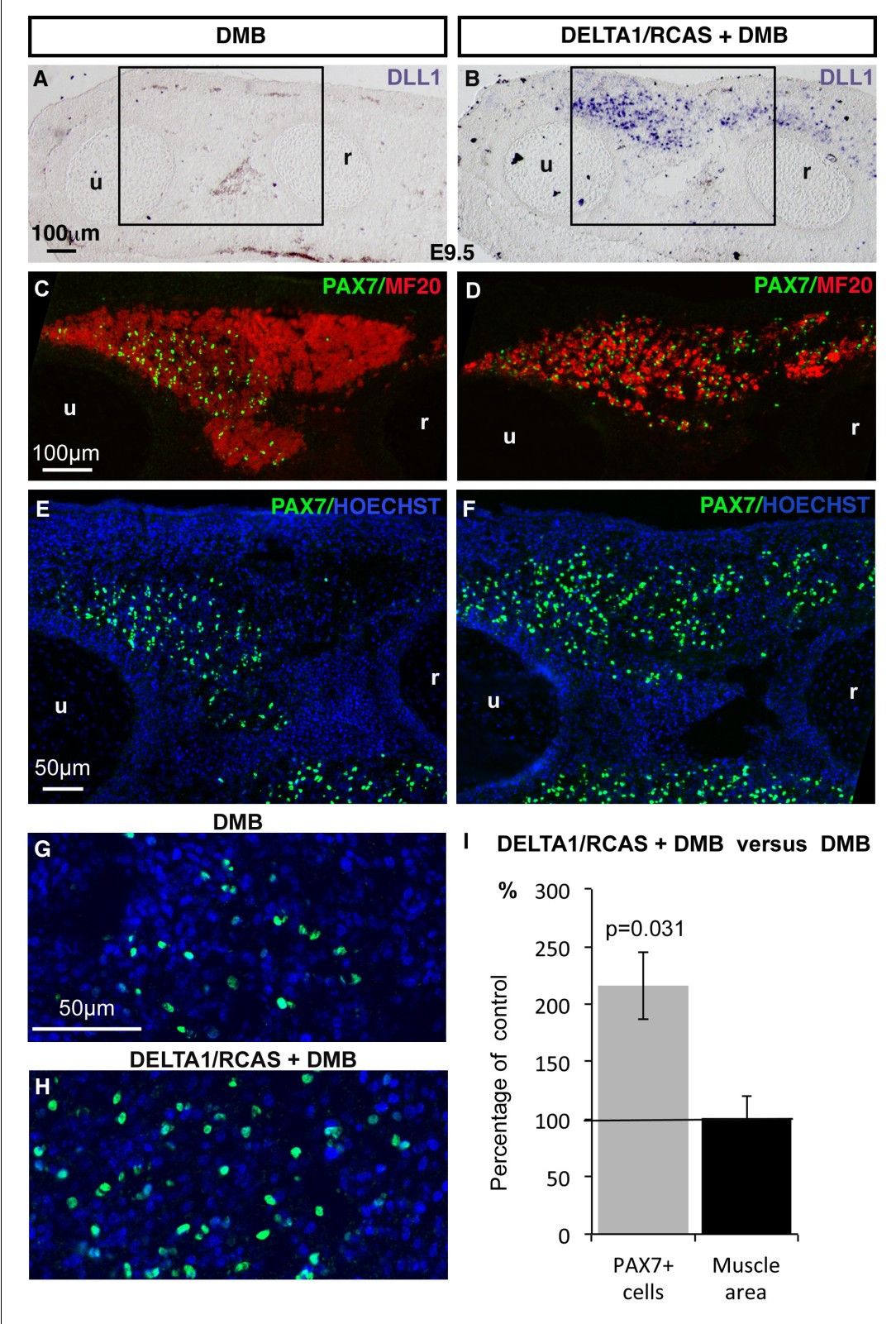

**Figure 4.** Ligand-induced NOTCH activity prevents the diminution in the number of fetal muscle progenitors and the increase of muscle differentiation in immobilized fetuses. Transverse sections of contralateral (A,C,E,G) and DELTA1/RCAS grafted (B,D,F,H) limbs of DMB-treated fetuses were hybridized with the DLL1 probe (A,B) to visualize ectopic *DLL1* expression in right grafted limbs (B) or immunostained with PAX7 and MF20 antibodies (C–H) (N = 3). (I) The quantification of PAX7+ cell number per unit area and muscle area was performed in DELTA1/RCAS-grafted and contralateral

*Figure 4 continued on next page*

*Figure 4 continued*

limbs of the same immobilized embryos. Result was presented as percentage of contralateral limbs, in immobilization conditions. Error bars represent standard deviations of six sections originating from three independent experimental embryos. The p-value was calculated using the Wilcoxon test. Limb sections are dorsal to the top and posterior to the left. u, ulna, r, radius.
The following figure supplement is available for figure 4:

**Figure supplement 1.** A continuous source of NOTCH ligand maintained the number of PAX7+ muscle progenitors despite the inhibition of muscle differentiation.

differentiated muscle cells activated *ANKRD1* expression in paralyzed muscles compared to the loss of ANKRD1 in paralyzed muscles (*Figure 7A,B,D,E*). This shows that YapS112A activates YAP target gene expression in paralyzed muscles. YAP forced-activity also rescued *JAG2* expression in paralyzed muscles compared to the loss of *JAG2* in paralyzed muscles (*Figure 7A,C,D,F*). Moreover, the number of PAX7+ cells was increased of 55.19% (± 10.2) in muscles displaying YapS112A expression in differentiated muscle cells (visualized with Tomato expression) compared to muscles of contralateral limbs (*Figure 7G–N*). We conclude that YAP forced-activity in muscle fibers prevents the loss of *JAG2* expression in muscle fibers and the decrease in the number of adjacent muscle progenitors in immobilization conditions.

## YAP is recruited to *JAG2* promoter in chick limb muscles

To define whether YAP directly regulates *JAG2* transcription, we performed in vivo ChIP experiments using chick limbs as chromatin source. ChIP sequencing data on chick limb cells analyzing promoter-associated histone marks allowed us to characterize the promoter of the chick *JAG2* gene (*Figure 8—figure supplement 1*). We identified a putative regulatory region (−629 bp; −1023 bp) in the *JAG2* promoter, which contained a MCAT element (CATTCC), the known binding motif for TEAD complexes (*Davidson et al., 1988*). We found that YAP was recruited to this regulatory region based on PCR (*Figure 8B*) and RT-q-PCR (*Figure 8D*) analyses. Further sequences, upstream of the *JAG2* transcription initiation site, containing (−8481 −8960 bp) or not containing (−4500 bp −4894 bp) MCAT elements were not occupied by YAP (*Figure 8A,B*). This result showed that YAP was recruited to the *JAG2* promoter region in limb fetal skeletal muscles. The YAP occupancy to this region was decreased in immobilized fetuses based on PCR (*Figure 8C*) and RT-q-PCR (*Figure 8D*) analyses, consistent with the decrease of *JAG2* expression in muscle fibers in immobilization conditions (*Figure 2*). The YAP recruitment to *JAG2* promoter (containing MCAT elements) provides a possible mechanism for the contraction-dependent control of *JAG2* expression in fetal muscle fibers.

## Discussion

### Muscle contraction maintains the fetal muscle progenitor pool by preventing differentiation

The influence of muscle contraction has been extensively studied for skeleton (reviewed in *Schwartz et al. (2013)* and *Shea et al. (2015)*) but not for skeletal muscle development. We show here that mechanical signals are required to maintain the PAX7+ muscle progenitor pool during fetal myogenesis. However, muscle contraction does not change cell proliferation but rather affects the maintenance of PAX7+ cells by preventing their differentiation. Although chondrocyte proliferation is reduced in growth plates of long bone in the absence of muscle contraction (*Roddy et al., 2011*), skeletogenesis is also affected by proliferation-independent mechanisms after immobilization. Muscle contraction is necessary to maintain progenitors of the joint in an undifferentiated state and prevents their differentiation into chondrocytes (*Kahn et al., 2009*). Muscle contraction also controls chondrocyte convergence extension during cartilage development in zebrafish and mice (*Shwartz et al., 2012*). This indicates that progenitor cells of the musculoskeletal system are sensitive to mechanical signals generated by muscle contraction during development and respond to mechanical activity by several mechanisms. Interestingly in plants, physical forces contribute to stem

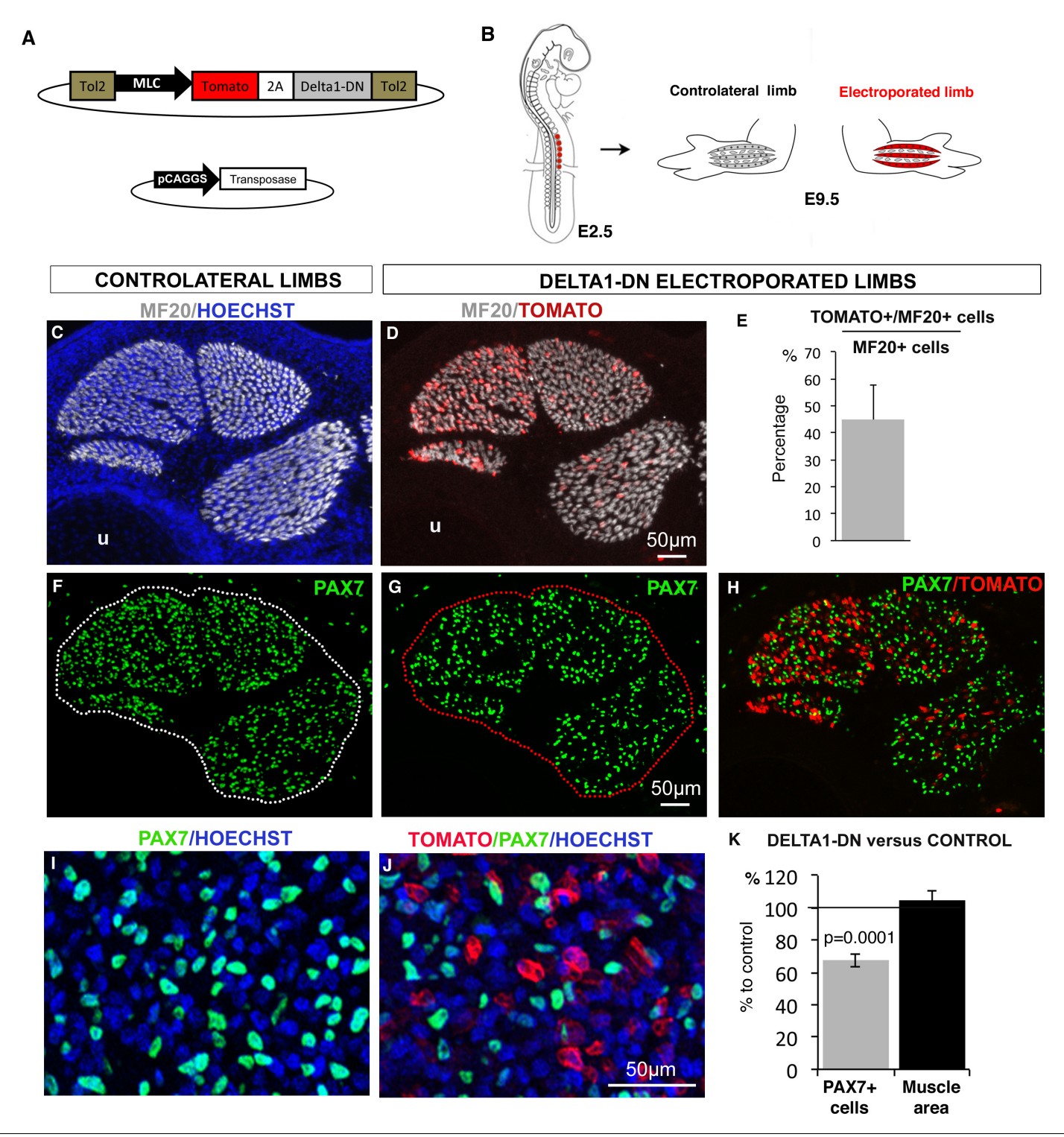

**Figure 5.** NOTCH-ligand activity in differentiated muscle cells is required to maintain the pool of fetal muscle progenitors. (**A**) Schematic representation of the recombinant vector co-expressing the Tomato reporter gene and a dominant-negative form of DELTA1 (DELTA1/DN) under the control of the Myosin Light Chain (MLC) promoter between two Tol2 transposons, and of the transient vector containing the transposase. (**B**) E2.5 chick embryos were electroporated at the level of forelimb somites in order to target limb muscle cells. Electroporated and contralateral limbs from the same animals (N = 4) were analyzed 7 days after electroporation, at E9.5. (**C,D, F–J**) Transverse sections of electroporated and contralateral limbs were immunostained with PAX7 and MF20 antibodies and labeled with Hoechst to visualize nuclei in blue. (**D,E**) In electroporated muscles displaying Tomato expression (**D**), an average of 44.94% (±12.88) of MF20+ cells displayed Tomato expression (**E**). The size and shape of the electroporated muscles was

*Figure 5 continued on next page*

*Figure 5 continued*

not affected (D) compared to control muscles (C). Electroporated muscles displayed a decrease in the number of PAX7+ cells (G,H,J) compared to contralateral limbs (F,I). (K) The number of PAX7+ cells per unit area and the muscle area were analyzed in electroporated muscles and equivalent muscles of the contralateral limbs (originating from the same experimental animals). Results were presented as percentage of the contralateral limbs (control). The graph shows means ± standard errors of the mean of 14 sections originating from four electroporated embryos. The p-value was calculated using the Wilcoxon test. Limb sections are dorsal to the top and posterior to the left. u, ulna.

cell maintenance acting on a master regulator (STM homeobox) of Arabidopsis shoot meristems, and this is achieved by a mechanism that is independent of cell proliferation (*Landrein et al., 2015*).

In the adult, changes in mechanical loading are known to cause variation of muscle size, but the contribution of satellite cells (adult muscle stem cells) to muscle atrophy or hypertrophy is a debated issue [reviewed in *Brooks and Myburgh (2014)*]. However, a reduction in the number of satellite cells has been described in adult muscle disuse/unloading animal models (*Mitchell and Pavlath, 2004*; *Verdijk et al., 2012*). Conversely, training exercises result in an increase of satellite cell number in humans (*Crameri et al., 2004*, *2007*; *Suetta et al., 2013*). Moreover, physiological exercise has been shown to increases satellite cell numbers and to be beneficial for sarcopenic muscles [reviewed in *Arthur and Cooley (2012)*]. Thus, our data and the available information in literature indicate that muscle stem cells are sensitive to mechanical forces during muscle development, homeostasis and ageing. It remains to be determined whether similar molecular or cellular mechanisms are active in these diverse settings.

## NOTCH signaling pathway acts downstream of mechanical signals during fetal myogenesis

Muscle paralysis causes a reduction of NOTCH activity and consequently muscle changes that are similar to those observed in NOTCH loss-of-function experiments. This shows that NOTCH activity is sensitive to mechanical signals during developmental myogenesis. Furthermore, ligand-induced NOTCH activation prevents the concomitant reduction of the fetal progenitor pool and increase muscle differentiation that is observed in the absence of muscle contraction. NOTCH components are also downregulated in bones of muscleless limbs of *Spd* mutant mice (*Rolfe et al., 2014*), although there is no described NOTCH loss-of-function phenotype in bones of muscleless limbs or immobilized embryos. NOTCH signaling is known to control the development of the cardiovascular system, which experiences mechanical forces. The NOTCH pathway has been proposed as an intermediary between mechanical forces and heart development [reviewed in *Granados-Riveron and Brook (2012)*], although the precise mechanotransduction pathway has not been identified. In adult muscle, NOTCH signaling has been extensively studied during muscle homeostasis, ageing and regeneration [reviewed in *Mourikis and Tajbakhsh (2014)*], and correlations between mechanical forces and NOTCH have been established (*Carlson et al., 2009*; *Conboy et al., 2003*). In summary, there is evidence for a change of NOTCH activity in different tissues during development and adult life upon muscle loading, but a functional link between mechanical signals and NOTCH has not been reported. We provide evidence that NOTCH signaling emerges as a molecular pathway that acts downstream of mechanical forces to regulate the pool of muscle progenitors during development. We believe that muscle activity acts on NOTCH ligand expression in fetal muscle fibers, since *JAG2* expression is lost in fibers upon unloading and the blockade of NOTCH ligand function in muscle fibers suffices to decrease the number of adjacent muscle progenitors. In zebrafish, swimming-induced exercises lead to extensive transcriptional changes in fast muscles, including modification of the expression of NOTCH ligands, *Dll1*, *Jag1* and *Jag2* in addition to other NOTCH components (*Palstra et al., 2014*). Interestingly, on a molecular level, mechanical forces exerted by the signal-sending cell are required for ligand-induced NOTCH activation in a receiving cell that does not appear to sense forces directly (*Gordon et al., 2015*; *Wang and Ha, 2013*). Lastly, the NOTCH ligand Jagged1 rescues the Duchenne muscular dystrophy phenotype in zebrafish embryos and is upregulated in mildly affected dystrophin-deficient dogs (*Vieira et al., 2015*). All these data converge to the idea that mechanical forces provided by muscle contraction are sensed by NOTCH ligands in muscle fibers.

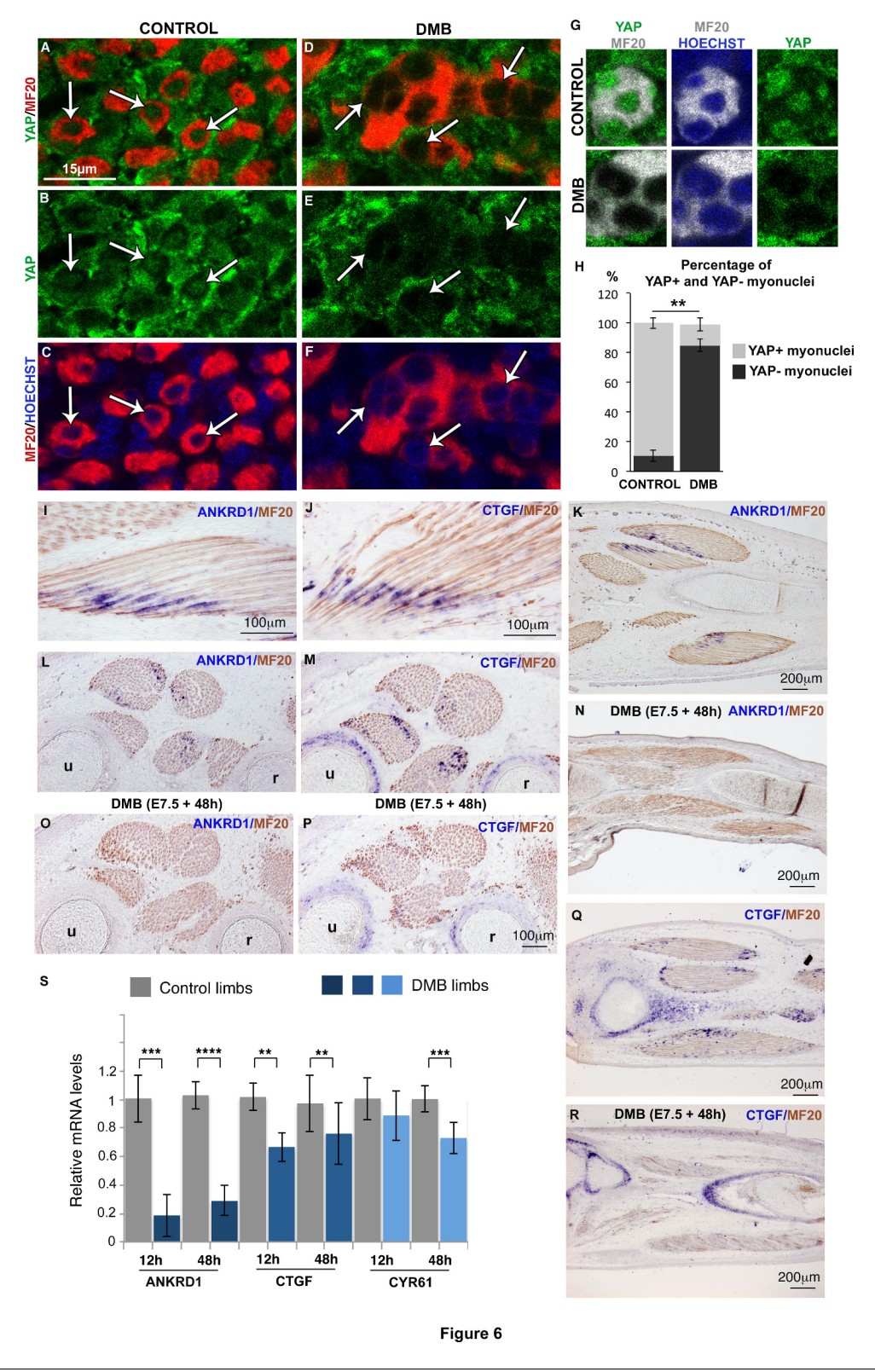

**Figure 6.** YAP activity is observed in contracting muscle fibers and lost in paralyzed muscles. (**A–F**) Transverse limb muscle sections of E9.5 control (**A–C**) and DMB-treated fetuses (**D–F**) were immunostained with YAP and MF20 antibodies and labeled with Hoechst to visualize nuclei in blue (N = 3). (**A–C**) In myofibers, YAP was preferentially localized in myonuclei (arrows). (**D–F**) In paralyzed limb muscles, the nuclear localization of YAP protein in muscle fibers (MF20+ cells in red) was lost (**D–F**, arrows) compared to control muscles (**A–C**, arrows). (**G**) Focus on YAP+ myonuclei in MF20+ cells in control

*Figure 6 continued on next page*

*Figure 6 continued*

muscles and YAP- myonuclei in paralyzed muscles (DMB). (**H**) Quantification of the percentage of YAP+ myonuclei in MF20+ cells in control and paralyzed muscles. In situ hybridization to YAP target genes, ANKRD1 (**I,K,L,N,O**) and CTGF (**J,M,Q,R**) followed by immunohistochemistry with MF20 antibody in control limbs (**I-M, Q**) and paralyzed muscles (**N,O,P,R**) (N = 4). (**I–M, Q**) The YAP target genes were expressed at the tips of muscle fibers (blue staining in MF20+ cells in brown) visualized on longitudinal (**I–K,Q**) and transverse (**L,M**) muscle sections. *ANKRD1* was exclusively expressed in limb muscles (**L,K**), while *CTGF* (**M,Q**) displayed additional expression in cartilage. (**N,O,P,R**) In the absence of muscle contraction, the expression of *ANKRD1* and *CTGF* was lost in muscles but not in cartilage for CTGF. u, ulna, r, radius. For transverse limb sections, dorsal is to the top and posterior to the left. For longitudinal sections, dorsal is to the top and proximal to the left. (**S**) RT-q-PCR analyses of the expression levels of YAP target genes in limbs of 12 hr and 48 hr DMB-treated embryos. For each gene, the mRNA levels of control limbs were normalized to 1. The graph shows means ± standard errors of the mean of nine biological replicates. The p-values were calculated using the Mann-Whitney test. Asterisks indicate the p-value, **p<0.01, ***p<0.001, ****p<0.0001.

The following figure supplement is available for figure 6:

**Figure supplement 1.** YAP expression and activity in limb muscles of control and immobilized fetuses.

## The molecular sensor of mechanical forces YAP regulates *JAG2* transcription in post-mitotic muscle fibers

We show here that YAP activity is sensitive to mechanical stimuli in fetal muscles in vivo. The preferential loss of YAP protein in myonuclei and YAP target gene expression in differentiated muscle cells in immobilized fetuses indicate that muscle fibers directly sense mechanical signals. Our results do not provide evidence that muscle progenitors directly sense mechanical signals, since nuclear YAP protein, proliferation rate and apoptosis were not modified in PAX7+ cells in immobilization conditions. Moreover, within fetal muscles, we did not detect any expression of the YAP target genes, *ANKRD1* and *CTGF* elsewhere than in MF20+ muscle fibers. YAP is, however, known to be expressed in activated satellite cells and has been shown to enhance satellite cell proliferation and to inhibit their differentiation in culture (*Judson et al., 2012*; *Watt et al., 2010*). Consistently, YAP is upregulated in alveolar rhabdomyosarcoma and is a potent modulator of rhabdomyosarcoma formation (*Crose et al., 2014*; *Tremblay et al., 2014*). However, in addition to YAP function in muscle cell proliferation, YAP was recently shown to be upregulated in muscle fibers by mechanical overload and to be critical for the size of skeletal muscle fibers in adult mice (*Goodman et al., 2015*; *Watt et al., 2015*), highlighting an additional YAP function in post-mitotic muscle fibers upon loading. YAP and NOTCH pathways have been shown to converge in many biological systems in mice (*Li et al., 2012*, *Yimlamai et al., 2014*, *Manderfield et al., 2015*). Oncogenic YAP variants activate the NOTCH ligand *JAG1* and consequently NOTCH signaling in hepatocellular carcinoma cells (*Tschaharganeh et al., 2013*). Moreover, TEAD binds to regulatory sequences of the *JAG1* gene in human breast cancer cells (*Zhao et al., 2008*). The current view is that YAP and NOTCH pathways converge to promote proliferation, cell fate and differentiation in a cell-autonomous manner in vertebrates. Using muscle unloading during fetal myogenesis, we provide evidence that YAP acts on muscle progenitors in a NOTCH–dependent and non-cell autonomous manner. This mechanism is reminiscent of that of the Drosophila YAP equivalent, Yorkie, during crystal cell differentiation in hematopoiesis, where Yorkie activates Serrate to induce responses in neighboring cells (*Ferguson and Martinez-Agosto, 2014*). We demonstrate a mechanistic link between YAP activity in muscle fibers and the regulation of the muscle progenitor pool and show that this is mediated by a transcriptional regulation of the NOTCH ligand *JAG2* (*Figure 9*). It remains to be determined if a similar link exists between YAP and NOTCH ligands in adult muscle fibers that could explain changes in satellite cell number during unloading or uploading of muscles. We believe that the molecular mechanism that we identify during fetal development might be central to control muscle maintenance upon mechanical loading.

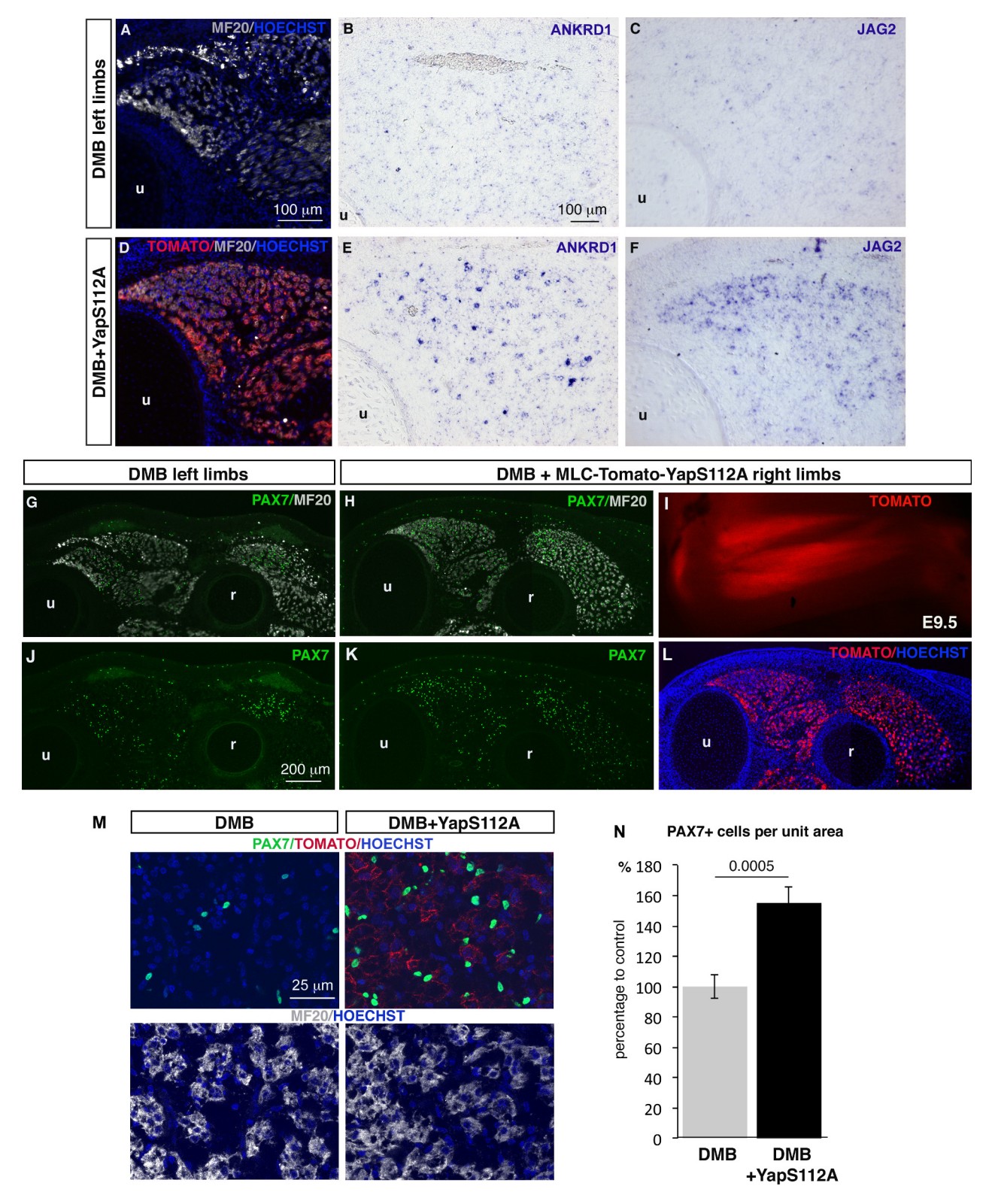

**Figure 7.** Forced YAP activity in differentiated muscle fibers prevents the decrease of *ANKRD1* and *JAG2* expression and in the number of fetal muscle progenitors in immobilized fetuses. E2.5 chick embryos were electroporated at the level of forelimb somites with MLC-Tomato-YapS112A to force YAP activity in differentiated muscle cells, and then treated with DMB. Contralateral and electroporated limbs from the same immobilized animals (N = 4) were analyzed 7 days after electroporation, at E9.5. (A–F) Adjacent sections of contralateral (A–C) and electroporated (D–F) limbs were immunostained

*Figure 7 continued on next page*

*Figure 7 continued*

with MF20 and labeled with Hoechst to visualize nuclei in blue (**A,D**) or hybridized with ANKRD1 (**B,E**) and JAG2 (**C,F**) probes. (**D**) Tomato indicates the electroporated muscle fibers. (**A–F**) *ANKRD1* (**E**) and *JAG2* (**F**) expression was activated in paralyzed muscles in the presence of YapS112A (**D**) in differentiated muscle cells compared to paralyzed muscles (**A–C**). (**I**) Tomato expression in wholemount MLC-Tomato-YapS112A-electroporated limbs. (**G,H,J–L**) Transverse sections of contralateral left (**G,J**) and MLC-Tomato-YapS112A electroporated right (**H,K,L**) limbs of DMB-treated fetuses were co-immunostained with PAX7 and MF20 antibodies. (**L**) Tomato expression in the same limb section as (**H,K**) to visualize electroporated muscle fibers. (**M**) High magnifications of dorsal muscle areas of DMB and DMB+YapS112A fetuses showing PAX7/TOMATO/HOECHST and MF20/HOECHST. (**N**) PAX7+ cell number was counted per unit area in muscles displaying Tomato expression in electroporated limbs and in equivalent muscles of contralateral limbs of four embryos. Results are shown as percentage of control. Error bars indicate standard deviations. The p-value was obtained using the Mann-Withney test. Limb sections are dorsal to the top and posterior to the left. u, ulna, r, radius.

The following figure supplement is available for figure 7:

**Figure supplement 1.** Forced YAP activity increases *JAG2* expression in chick fetal myoblasts.

## Materials and methods

### Chick embryos

Fertilized chick eggs from commercial sources (White Leghorn, HAAS, Strasbourg, France and JA57 strain, Morizeau, Dangers, France) were incubated at 37.5°C. Chick embryos were staged according to days in ovo.

### Drug administration

Decamethonium bromide (DMB) (Sigma, France) and pancuronium bromide (PB) (Sigma) solutions were freshly prepared before each experiment at 12 mM or 11 mM, respectively, in Hank's solution (Sigma) with Penicillin-Streptomycin at 1% (Gibco, France). The control solution was prepared using Hank's solution with 1% of Penicillin-Streptomycin. 100 µl of DMB, PB or control solutions were administrated in chick embryos at E7.5 and E8.5. Embryos were fixed at E8 for the 12 hr time point, at E8.5 for the 24 hr time point and at E9.5 for the 48 hr time point.

### Grafts of DELTA1-RCAS-expressing cells

Chick embryonic fibroblasts (CEFs) obtained from E10 chick embryos were transfected with DELTA1/RCAS using the Calcium Phosphate Transfection Kit (Invitrogen, France). Cell pellets of approximately 50–100 µm in diameter were grafted into limb buds of E3.5 embryos as previously described (*Bonnet et al., 2010*; *Delfini et al., 2000*). DELTA1/RCAS-grafted embryos were either harvested at E9.5 or treated with DMB or control solution and harvested at E9.5.

### Construction of electroporation vectors and electroporation

The pT2AL-MLC-Tomato-2A-DELTA1/DN was designed as follows: the dominant negative form of DELTA1 was defined as a truncated form of DELTA1 that lacks the intracellular domain (*Henrique et al., 1997*) and was amplified by PCR from the DELTA1/RCAS (*Delfini et al., 2000*) using the following primer sequences: Fw GACTTCGAAATGGGAGGCCGCT and Rv CACGTGTTAC TATCACCTGCAGGCCTCG. The DELTA1/DN PCR product was inserted in the pT2AL-MLC-Tomato-2A-GFP (*Bourgeois et al., 2015*) after GFP removal to obtain pT2AL-MLC-Tomato-2A-DELTA1/DN (named as pT2AL-MLC-Tomato-DELTA1/DN). The pT2AL-MLC-Tomato-2A-mYapS112A was designed as follows: mYapS112A (*Xin et al. 2013*) was amplified by PCR from the mYapS112A/ RCAS (*McKey et al., 2016*) using the following primer sequences: Fw CAATTCGAAA TGGAGCCCGCG and Rv GGCCACGTGCTATAACCACGTGAGAAA. The mYapS112A PCR product was inserted in the pT2AL-MLC-Tomato-2A-GFP (*Bourgeois et al., 2015*) after GFP removal to obtain pT2AL-MLC-Tomato-2A-mYapS112A (named as pT2AL-MLC-Tomato-YapS112A). Forelimb somite electroporation was performed as previously described (*Wang et al., 2011*). The DNA solution was composed of the pT2AL-MLC-Tomato-DELTA1/DN or pT2AL-MLC-Tomato-YapS112A vectors and a transient transposase-containing vector pCAGGS-T2TP, at a molar ratio of 3:1. This vector set allows the stable integration of the MLC-Tomato-DELTA1/DN or the MLC-Tomato-

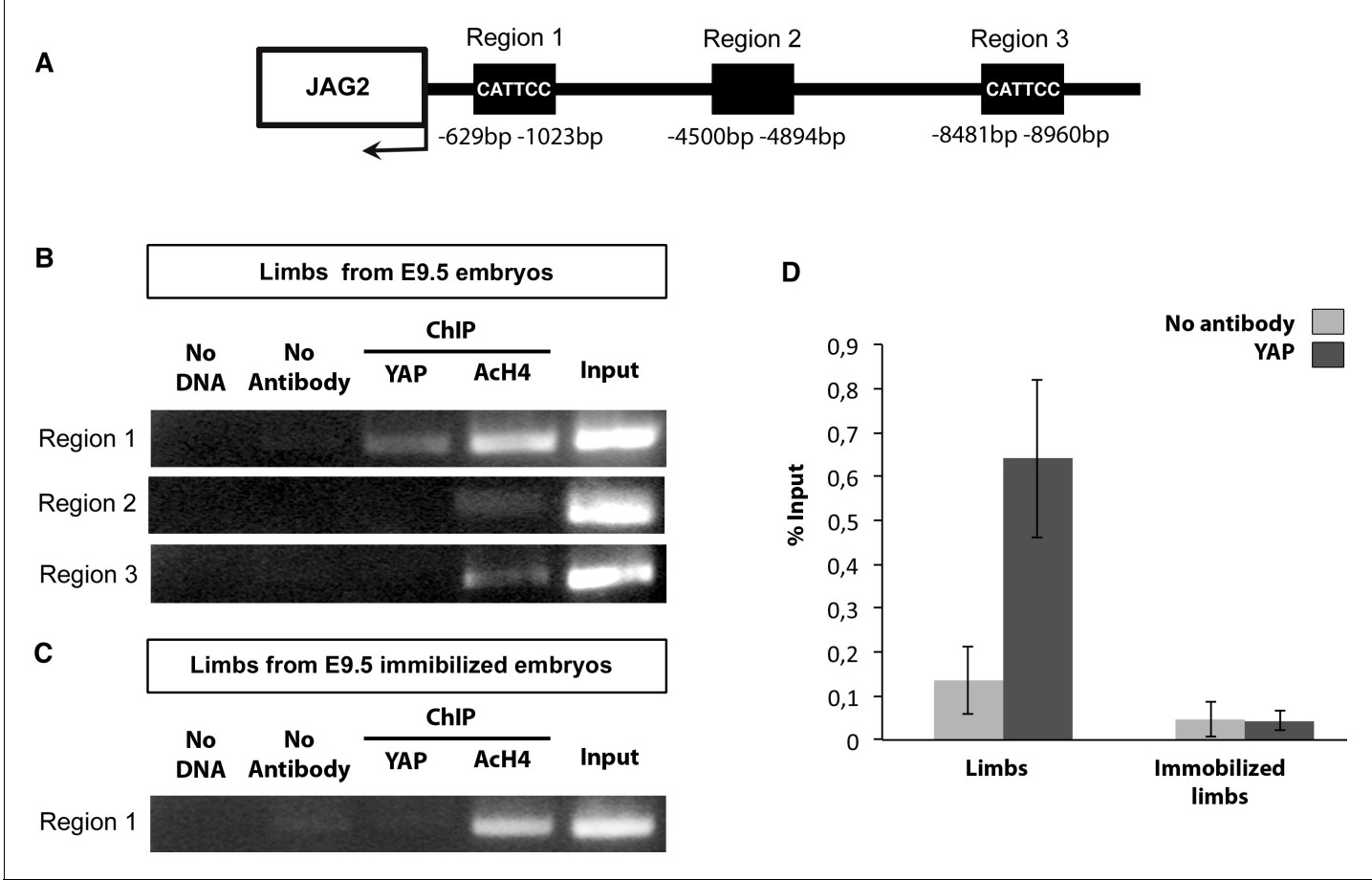

**Figure 8.** YAP is recruited to a MCAT element-containing promoter region of the chick *JAG2* gene in fetal muscles upon muscle contraction. ChIP assay was performed from eight limbs of E9.5 chick control or immobilized fetuses with antibodies against YAP, AcH4 for positive control, or without antibody as a negative control in two independent biological experiments. ChIP products were analyzed by PCR (**B**,**C**) (N = 4) or by RT-q-PCR (N = 2) (**D**). Primers targeting a 394 pb fragment named region 1 (−629 bp −1023 bp) in the *JAG2* promoter region (**A**) identified a DNA sequence immunoprecipitated by YAP by PCR (**B**) or by RT-q-PCR (**D**), while primers targeting regions 2 and 3 did not show any immunoprecipitation by PCR (**B**). (**D**) Experiment showing the signal of relative YAP recruitment to *JAG2* regulatory region 1 in control and immobilized limbs. Results were represented as percentage of the input. Error bars show standard deviations. The YAP recruitment to *JAG2* regulatory region 1 was lost in the absence of muscle contractions assessed by PCR (**C**) and RT-q-PCR (**D**) analyses.

The following figure supplement is available for figure 8:

**Figure supplement 1.** ChIP sequencing data with promoter histone marks on the chick *JAG2* gene.

YapS112A cassettes into the chick genome. Embryos electroporated with pT2AL-MLC-Tomato-YapS112A were immobilized with DMB treatment.

## Myoblast cultures

Primary myoblasts were obtained from hindlimbs of E10 chick embryos, as previously described (*Havis et al., 2012*). DMB and PB were applied to proliferating myoblast cultures in high-serum conditions (10%), at a final concentration of 50 µM and 5 µM, respectively, for 48 hr. Myoblasts were then analyzed by immunohistochemistry for PAX7+ muscle progenitors and MF20+ differentiated cells and for muscle gene expression by RT-q-PCR. To force YAP activity in muscle cells, myoblasts were transfected with YapS112A/RCAS (*McKey et al., 2016*). Muscle cells were amplified in high-serum conditions until confluence, collected and analyzed for gene expression by RT-q-PCR. To obtain differentiated muscle cells that overexpress YapS112A, chick fetal myoblasts were transfected with pT2AL-MLC-Tomato-YapS112A or pT2AL-MLC-Tomato as control and cultured for 4 days. Cells

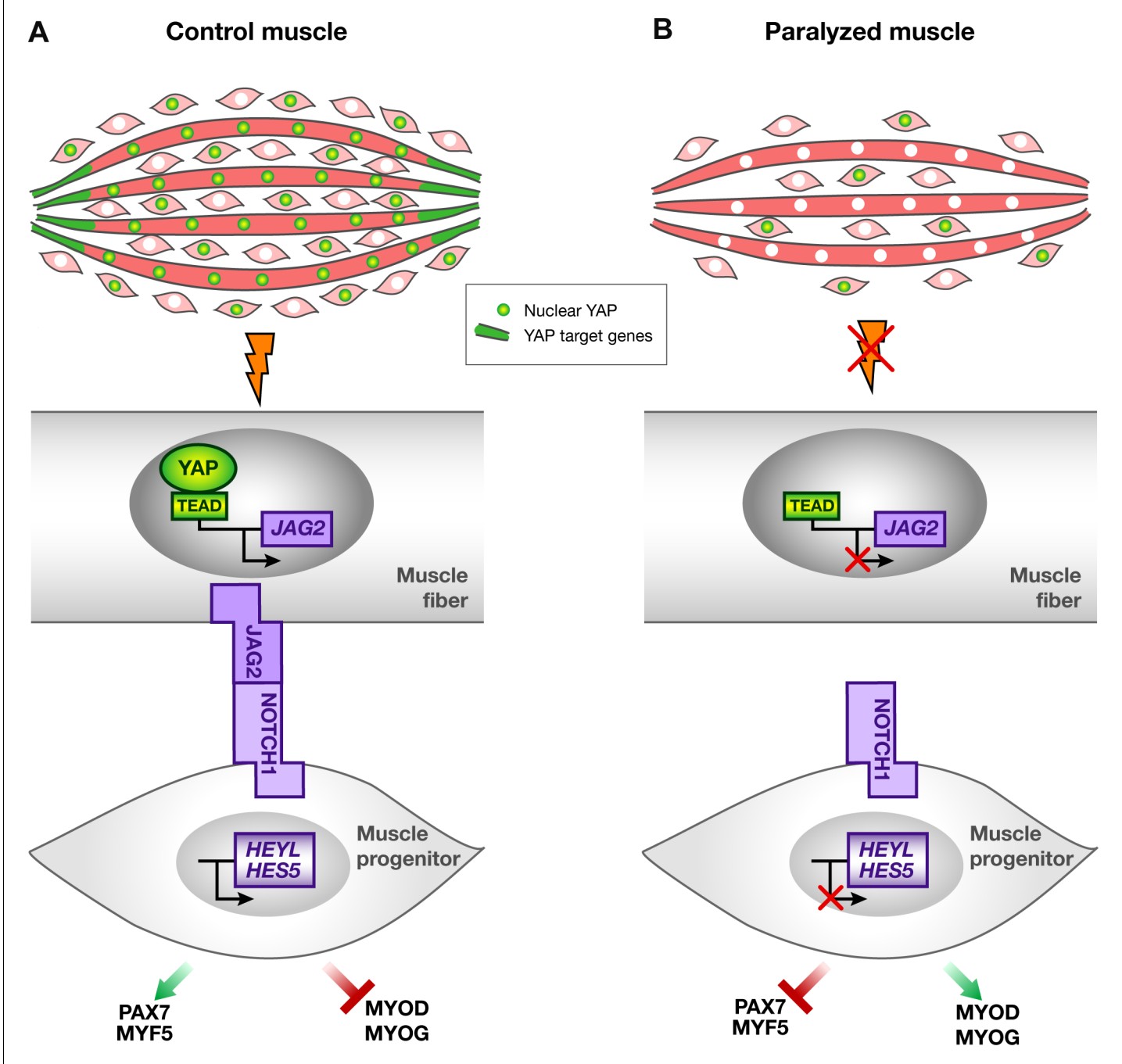

**Figure 9.** Schematic representation of YAP and NOTCH signaling pathways in normal contracting muscles and paralyzed muscles. (**A**) In contracting muscles, nuclear YAP (green myonuclei) and YAP target gene transcripts (green) are present in post-mitotic muscle fibers. YAP positively regulates *JAG2* transcription upon muscle contraction. Ligand-dependent NOTCH activation regulates the muscle progenitor pool, by preventing muscle progenitors to differentiate. (**B**) In paralyzed muscles, nuclear YAP, YAP target genes and *JAG2* transcripts are lost in post-mitotic muscle fibers. The absence of the NOTCH ligand JAG2 in fibers, due to the loss of mechanical signals, induces a NOTCH loss-of-function phenotype i.e. a diminution in the number of muscle progenitors and a shift toward differentiation, in a non-cell autonomous manner.

were collected into PBS 1X 10% fetal bovine serum (FBS), stained with DAPI to exclude dead cells and purified via FACS Aria gating on the Tomato+ cell fraction.

## RNA extraction and quantitative real-time PCR

Total RNAs were extracted from control limbs, experimental limbs or primary fetal myoblast cultures. 500 ng to 1 μg of RNA was reverse-transcribed using the High-Capacity Retrotranscription kit (Applied Biosystems, France). RT-q-PCR was performed using SYBR Green PCR Master Mix (Applied Biosystems). Primer sequences used for RT-q-PCR are listed in *Supplementary file 1*. The relative mRNA levels were calculated using the $2^{-\Delta\Delta Ct}$ method (*Livak and Schmittgen, 2001*). The ΔCts were obtained from Ct normalized with GAPDH and RPS17 levels in each sample. Twelve DMB, PB or control limbs, originating from four independent experiments, were used as independent RNA samples. Six samples of cultured myoblasts originating from three independent experiments were used as independent RNA samples for DMB, PB, YapS112A/RCAS or pT2AL-MLC-Tomato-YapS112A experiments. Each RNA sample was analyzed in duplicate. Errors bars in RT-q-PCR results represent standard error of the mean.

## EdU experiments

Fifty microliters of EdU (Invitrogen) solution (5 mg/ml) was injected into the circulation in E9.5 chick embryos for 1.5 hr. Embryos were then fixed and immunohistochemistry was performed on limb transverse sections.

## Immunohistochemistry

Forelimbs of control or manipulated (DMB, PB, DELTA1/RCAS, DELTA1/RCAS+DMB, pT2AL-MLC-Tomato-DELTA1/DN, pT2AL-MLC-Tomato-YapS112A+DMB) chick embryos were fixed in 4% paraformaldehyde overnight at 4°C and then processed in gelatin/sucrose for 12 μm cryostat sections. The monoclonal antibodies, MF20 that recognizes sarcomeric myosin heavy chains and PAX7 that recognizes muscle progenitors, developed by D.A. Fischman and A. Kawakami, respectively, were obtained from the Developmental Studies Hybridoma Bank developed under the auspices of the NICHD and maintained by The University of Iowa, Department of Biology Iowa City, IA 52242. The Desmin antibody was obtained from Sigma (dilution 1/100). The YAP mouse monoclonal antibody was obtained from Santa Cruz Biotechnology (dilution 1/50). The MEP21 antibody originates from (*McNagny et al., 1997*). Proliferation analysis (EdU) was performed using the Click-iT kit (Thermo Fisher Scientific, France). Apoptosis was detected using the ApoTag kit (Millipore, France). Secondary antibodies were conjugated with Alexa-488 or Alexa-555 (Invitrogen). Nuclei were detected with Hoechst staining (Molecular Probes).

## In situ hybridization

Chick forelimbs of control or manipulated (DMB, PB, DELTA1/RCAS, DELTA1/RCAS+DMB, pT2AL-MLC-Tomato-DELTA1/DN, pT2AL-MLC-Tomato-YapS112A+DMB) embryos were fixed in Farnoy (60% ethanol, 30% formaldehyde (stock at 37%) and 10% acetic acid) overnight at 4°C, and processed for in situ hybridization on wax tissue sections, as previously described (*Wang et al., 2010*). The digoxigenin-labeled mRNA probes were obtained as follows: DLL1, JAG2 and MYOD (*Delfini et al., 2000*); HES5 (*Henrique et al., 1997*). CTGF (EST clones) and YAP1 (*McKey et al., 2016*) probes were linearized with NotI and SacII and synthetized with T3 and SP6, respectively. ANKRD1 probe was obtained by PCR from E9.5 limb tissues using the following primers: Fw: TGGC TCACGGGAAGGAGAAG; Rv: GGTGCTCGGCACAGTCG, cloned into the pCRII-TOPO vector (Invitrogen), linearized with KpnI and synthetized with T7.

## Chromatin immunoprecipitation assay

ChIP assay was performed as previously described (*Havis et al., 2012*). Eight limbs from E9.5 chick embryos were homogenized using a mechanical disruption device (Lysing Matrix A, Fast Prep MP1, 40 s at 6 m/s). 10 μg of the YAP rabbit polyclonal antibody (Santa Cruz Biotechnology) or 10 μg of the anti-acetylated histone H4 (AcH4) antibody (Upstate Biotechnology) was used to immunoprecipitate 20 μg of sonicated chromatin. ChIP products were analyzed by PCR to amplify three regions upstream the *JAG2* coding sequence or by RT-q-PCR to amplify the region 1 (*Figure 8A*). The

primer list is displayed in *Supplementary file 1*. ChIP assay was performed with or without DMB treatment in two independent biological replicates.

### Image capturing

After immunohistochemistry or in situ hybridization experiments, images were obtained using a Nikon epifluorescence microscope, a Leica DMI600B fluorescence microscope or a Leica SP5 confocal system.

### Cell number and muscle area measurements

All cell number and muscle area measurements were performed using the free software ImageJ (Rasband, W.S., ImageJ, U. S. National Institutes of Health, Bethesda, Maryland, USA, http://imagej.nih.gov/ij/, 1997–2012). To quantify the number of PAX7+ cells in limb muscles of DMB- and PB-treated fetuses, the number of PAX7+ cells was counted per unit area in dorsal and ventral muscles on three sections of each three different limbs originating from either DMB-treated, PB- treated or control embryos. The quantification of PAX7+ cells and muscle area (delineated with PAX7 and MF20 expression domains) in DELTA1/RCAS-grafted embryos was performed in three sections of each grafted (right) and contralateral (left) limbs of three immobilized and three control embryos. Quantification was performed in ectopic DLL1-expressing muscles of grafted right limbs and compared with equivalent muscles of the contralateral left limbs originating from the same embryos. The quantification of the Tomato+/MF20+ cells, MF20+ cells, PAX7+ cells and muscle area in pT2AL-MLC-Tomato-DELTA1/DN electroporated embryos was performed in five sections of each electroporated (right) and contralateral (left) of four embryos. Quantification was performed in Tomato-expressing muscles of electroporated right limbs and compared with equivalent muscles of the contralateral left limbs originating from the same embryos. The quantification of PAX7+ cells in pT2AL-MLC-Tomato-YapS112A electroporated and immobilized fetuses was performed in four sections of each electroporated (right) and contralateral (left) of four immobilized fetuses. Quantification was performed in Tomato-expressing muscles of electroporated right limbs and compared with equivalent muscles of the contralateral left limbs originating from the same embryos. Hoechst+ nuclei overlapping with *MYOD* expression were counted in four different muscles over four different sections and compared with the total number of Hoechst+ nuclei in paralyzed (DMB 48 hrs) and control muscles.

### Statistical analyses

Data were analyzed using non-parametric two-tailed tests, Mann-Withney test for unpaired samples or Wilcoxon test for paired samples using Graphpad Prism V6. Results were shown as means ± standard deviations or standard errors of the mean, depending on the size of the samples. The p-values are indicated either with the number on the graphs or with asterisks. Asterisks indicate the different p-values *p<0.05, **p<0.01 and ***p<0.001.

## Acknowledgements

We thank Pascal de Santa Barbara (Montpellier, France) and Thierry Jaffredo (Paris, France) for reagents, Laurence Petit (Paris, France) for help with FACS sorting. We thank laboratory members for comments on the manuscript and Sophie Gournet for illustrations.

## Additional information

### Funding

| Funder | Grant reference number | Author |
| --- | --- | --- |
| Agence Nationale de la Recherche | ANR-12-BSV1-0038 | Delphine Duprez |
| AFM-Téléthon | AFM 16752/16826 | Delphine Duprez |
| Fondation pour la Recherche Médicale | DEQ20140329500 | Delphine Duprez |

| Centre National de la Recherche Scientifique | Delphine Duprez |
| --- | --- |
| Institut National de la Santé et de la Recherche Médicale | Delphine Duprez |
| Université Pierre et Marie Curie | Delphine Duprez |

The funders had no role in study design, data collection and interpretation, or the decision to submit the work for publication.

## Author contributions
JEdL, Performed and analyzed experiments, Drafted the manuscript; M-AB, Performed experiments; CB, Analysed the data, Wrote the manuscript; DD, Designed experiments, Analyzed data, Wrote the manuscript, Acquisition of data

## Author ORCIDs
Joana Esteves de Lima, http://orcid.org/0000-0002-5497-5669
Delphine Duprez, http://orcid.org/0000-0003-0248-7417

## Additional files

### Supplementary files
• Supplementary file 1. List of primers used for RT-q-PCR analyses and ChIP assays.

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
