## [Decision Letter]

Thank you for submitting your article "Muscle contraction is required to maintain the pool of muscle progenitors via YAP and NOTCH during fetal myogenesis" for consideration by *eLife*. Your article has been reviewed by two peer reviewers, and the evaluation has been overseen by a Reviewing Editor and Fiona Watt as the Senior Editor. The following individual involved in review of your submission has agreed to reveal her identity: Gabrielle Kardon (Reviewer #2).

The reviewers have discussed the reviews with one another and the Reviewing Editor has drafted this decision to help you prepare a revised submission.

The regulation of the pool of muscle progenitors during development is a critical determinant of muscle size. In this paper, the authors test the novel hypothesis that muscle contraction is important for regulating the number of muscle progenitors during development. By using drugs that block the contraction of muscles in the chick embryo and by molecular manipulations, they describe the possible role that YAP and NOTCH signaling may play in this process. Altogether this is a nice study with interesting hypotheses put forward which have potentially important implications, however a number of important points need to be addressed before publication can be envisaged.

1) In the Introduction, presentation of the role of the Notch pathway in myogenesis is inadequate. This is a complex situation (see for instance Mourikis and Tajbakhsh in 2014 for a partial outlook on this complexity) and presenting it as linear and simple is bound to attract criticism. The authors only refer to the role of Notch in the inhibition of myogenic differentiation of resident progenitors and satellite cells. However, opposite results have been described, e.g. in the Notch3 null mutation or the early differentiation of progenitors from the dermomyotome in the chick embryo (Kitamoto and Hanaoka, 2010; Rios et al., 2011). What this variety of responses means is that during myogenesis, Notch has different roles depending on the cellular and molecular context where it acts. This is something that should be clearly presented so that the text is less dogmatic from the start. As it is, it seems that studies that don't fit to the general hypothesis developed here are just not mentioned. In general, there is a bias towards citing much of the authors and co-authors work (and views) on what Notch is doing during myogenesis and missing other important references not only when they do not fit to the picture (see above) but even when they support the general idea (e.g. Bjornson et al. 2012).

2) The authors show that there is an increase of MyoD or Myog expression after drug treatment, which is paralleled by a decrease of PAX7 expression. They draw the conclusion that PAX7-positive cells have undergone myogenic differentiation. However, there is no definite proof that this is the case. The number of MyoD+ (or Myog+) cells could be the same after treatment, but with the number of PAX7+ cells decreasing, the proportion of MyoD+ or Myog+ cells would automatically increase (in particularly since the PAX7+ population is large). A more direct proof that this a differentiation process would be to demonstrate that the number of PAX7+ cells (per area unit) at t=0 is similar to that of MyoD+ cells per unit area at t=12 or 48 hrs. Since the authors claim there is no apoptosis (see below) the numbers should match. Furthermore a time course on MyoD and PAX7-positive cells would help.

3) The authors show that there is no change in EdU labelling in the PAX7+ population, and they conclude that there is no change in the proliferation of resident progenitors, thus supporting the hypothesis of an increase in differentiation (see above). However, the authors show that there is an increase of apoptosis in PAX7-negative cells in the muscle masses. How can they exclude that after drug treatment, the resident progenitors lose PAX7 expression, then undergo apoptosis? It is a concern that the cell population undergoing apoptosis is not identified.

4) The Notch ligand Jag2 and the Notch response (Hes5 and HeyL) are more strongly reduced after 12 than 48h, while the phenotype is stronger after 48h (Figure 1). The authors should discuss an explanation for this time difference.

5) The over-expression of Dll1 with RCAS viruses will target both the resident progenitors and the differentiated fibers. Since it is known that Notch is activated in trans and inhibited in cis (see del Alamo et al. Curr. Biol. 2011 for review), what conclusion can be drawn from this experiment? It would be important in these overexpression experiments to check cell type specificity, using the available tools.

In addition, since the authors claim it is Jag2 that is the ligand, why not use its cDNA? They could have used electroporation of Jag2 in combination with the muscle-specific promoter they used with the next experiment (DN Dll1) that would drive its expression in differentiated fibers. It would also have been appropriate to use siRNA directed against Jag2. The reviewers do not insist on redoing the experiments with Jag2, although if the authors have these data they should be included. in any case, since Jag2 is membrane bound, they should show that they have an effect only in cells in direct contact with the Jag2/ Dll1 over-expressing fibers.

6) It is not clear that Yap activation is downstream of muscle contraction and also that Yap activates Jag2 expression in myofibers. Of course, the most convincing data would be for the authors to show that ectopic activation of YAP in myofibers (via Tol2 MLC electroporation) rescues the loss of PAX7+ progenitors when myofibers have been immobilized. This experiment would considerably strengthen the paper and should be do-able by the authors since they have the Tol2 MLC construct.

7) Since YAP is such an ubiquitously expressed molecule, the authors should be careful in the phrasing of the conclusions.

---

## [Author Response]

*1) In the Introduction, presentation of the role of the Notch pathway in myogenesis is inadequate. This is a complex situation (see for instance Mourikis and Tajbakhsh in 2014 for a partial outlook on this complexity) and presenting it as linear and simple is bound to attract criticism. The authors only refer to the role of Notch in the inhibition of myogenic differentiation of resident progenitors and satellite cells. However, opposite results have been described, e.g. in the Notch3 null mutation or the early differentiation of progenitors from the dermomyotome in the chick embryo (Kitamoto and Hanaoka, 2010; Rios et al., 2011). What this variety of responses means is that during myogenesis, Notch has different roles depending on the cellular and molecular context where it acts. This is something that should be clearly presented so that the text is less dogmatic from the start. As it is, it seems that studies that don't fit to the general hypothesis developed here are just not mentioned. In general, there is a bias towards citing much of the authors and co-authors work (and views) on what Notch is doing during myogenesis and missing other important references not only when they do not fit to the picture (see above) but even when they support the general idea (e.g. Bjornson et al. 2012).*

In the original Introduction section describing the NOTCH pathway, we focused on limb foetal myogenesis, which involves the inhibition of muscle differentiation and the maintenance of muscle progenitors. We are aware that additional roles of NOTCH have been described during embryonic and adult myogenesis. We now mention the role of NOTCH during the activation of embryonic myogenesis in axial somites (Rios et al., 2011) and during activation, proliferation and quiescence of satellite cells. We also cite the Bjornson et al. (2012) paper, which shows a role for NOTCH in maintaining the number of satellite cells by inhibiting differentiation.

*2) The authors show that there is an increase of MyoD or Myog expression after drug treatment, which is paralleled by a decrease of PAX7 expression. They draw the conclusion that PAX7-positive cells have undergone myogenic differentiation. However, there is no definite proof that this is the case. The number of MyoD+ (or Myog+) cells could be the same after treatment, but with the number of PAX7+ cells decreasing, the proportion of MyoD+ or Myog+ cells would automatically increase (in particularly since the PAX7+ population is large). A more direct proof that this a differentiation process would be to demonstrate that the number of PAX7+ cells (per area unit) at t=0 is similar to that of MyoD+ cells per unit area at t=12 or 48 hrs. Since the authors claim there is no apoptosis (see below) the numbers should match. Furthermore a time course on MyoD and PAX7-positive cells would help.*

Unfortunately, there is no good commercial antibody against chick MYOD or MYOG (in contrast to the situation in mice), so we cannot calculate the number of PAX7+ and MYOD+ cells. In order to estimate the number of MYOD-expressing cells, we therefore counted the number of Hoechst+ nuclei of MYOD-expressing cells (based on MYOD in situ hybridization) versus the total number of Hoechst+ nuclei in muscles. We observed an increase in the percentage of MYOD-expressing cells in paralyzed muscles compared to control muscles (New Figure 1). Moreover, we observed larger Myosin+ fibres, with multiple grouped nuclei in paralyzed muscles compared to control muscles (New Figure 1). The increase in the number of MYOD-expressing cells, of MYOD and MYOG expression (in situ hybridization and RT-q-PCR) combined with the larger muscle fibres with multiple nuclei in paralyzed muscles compared to control muscles leads us to conclude that the decrease in the number of PAX7+ cells is due to increased differentiation. Lastly, the reversion of the muscle phenotype i.e. diminution of differentiation (and increase in the number of PAX7+ cells) upon DLL1- activated NOTCH in immobilisation conditions, is also an indirect argument that there is an increase of differentiation (in addition to the decrease in the number of PAX7+ cells) in immobilisation conditions (Figure 4).

We have added the 24 hour time point for PAX7, MYF5, MYOD and MYOG expression levels in immobilised limbs versus control limbs in Figure 1. This provides us with an intermediary time point between 12 hours and 48 hours.

*3) The authors show that there is no change in EdU labelling in the PAX7+ population, and they conclude that there is no change in the proliferation of resident progenitors, thus supporting the hypothesis of an increase in differentiation (see above). However, the authors show that there is an increase of apoptosis in PAX7-negative cells in the muscle masses. How can they exclude that after drug treatment, the resident progenitors lose PAX7 expression, then undergo apoptosis? It is a concern that the cell population undergoing apoptosis is not identified.*

We have looked at apoptosis 24 hours after DMB treatment and there is no increased apoptosis in paralysed muscles compared to control muscles. Since there are no double labeled PAX7+/Tunel+ cells 48 hours after immobilisation, we conclude that muscle progenitors do not undergo apoptosis in paralyzed muscles (48 hours after immobilisation).

The apoptosis observed in muscles paralyzed for 48 hours is rarely observed in MF20 + cells or Desmin+ cells (now added in Figure 3). Desmin expression allows us to cover myogenic cells between progenitors (PAX7) and fibres (MF20). This result shows that myogenic cells do not undergo apoptosis upon immobilisation. One likely hypothesis is that apoptosis is observed in non-myogenic cells, such as muscle connective tissue cells.

*4) The Notch ligand Jag2 and the Notch response (Hes5 and HeyL) are more strongly reduced after 12 than 48h, while the phenotype is stronger after 48h (Figure 1). The authors should discuss an explanation for this time difference.*

Since JAG2 is also expressed in vessels, it is difficult to speculate on a stronger effect at 12 h 4 than 48h. We believe that the q-PCR experiments confirm the decrease of JAG2 expression in limbs, which is clearly observed in limb muscles by in situ hybridisation (Figure 2).

*5) The over-expression of Dll1 with RCAS viruses will target both the resident progenitors and the differentiated fibers. Since it is known that Notch is activated in trans and inhibited in cis (see del Alamo et al. Curr. Biol. 2011 for review), what conclusion can be drawn from this experiment? It would be important in these overexpression experiments to check cell type specificity, using the available tools.*

DLL1 overexpression experiments with the RCAS virus lead to a NOTCH gain-of-function phenotype in chick limbs (Delfini et al., 2000, Bonnet et al., 2010, Havis et al., 2012) or in chick somites (Hirsinger et al., 2001). DLL1-activated NOTCH maintains the expression of muscle progenitor markers including PAX3, MYF5 and FGFR4, while inhibiting the expression of muscle differentiation genes, MYOD, MYOG, DES and MEF2C and Myosins (Bonnet et al., 2010, Havis et al., 2012). Based on the RCAS system, DLL1 will be overexpressed only in a subset of proliferating cells (any cell types). Consequently, DLL1 will act in trans to adjacent cells. Therefore, even if a subset of proliferating cells overexpressing DLL1 inhibited NOTCH in cis, the DLL1 signal is sufficient to activate

NOTCH in trans according to the phenotypes observed after DLL1 overexpression in muscles, as reported in the papers cited above.

*In addition, since the authors claim it is Jag2 that is the ligand, why not use its cDNA? They could have used electroporation of Jag2 in combination with the muscle-specific promoter they used with the next experiment (DN Dll1) that would drive its expression in differentiated fibers. It would also have been appropriate to use siRNA directed against Jag2. The reviewers do not insist on redoing the experiments with Jag2, although if the authors have these data they should be included.*

We used the DLL1/RCAS virus since it has been proven to induce consistently a NOTCH gain-of-function phenotype in limb muscles, i.e. an inhibition of muscle differentiation (based on the inhibition of differentiation muscle markers) and maintenance of progenitors (based on the expression of muscle progenitor markers) (Delfini et al., 2000; Bonnet et al., 2010; Havis et al., 2012). Moreover, we now show that the long-term activation of (DLL1-mediated) NOTCH maintains the number of PAX7+ cells despite the absence of differentiated muscle cells (Figure 4—figure supplement 1). This result is fully consistent with mouse data that show that activated Notch (NICD in Myf5+ cells) sustains muscle progenitors despite a block of differentiation (Mourikis et al., 2012).

To block NOTCH ligand activity, we have used an already described dominant negative form of DLL1, which has been shown to block the activity of all NOTCH ligands by preventing NOTCH ligand processing in signal-sending cells and therefore blocked NOTCH activation in signal-receiving cells (Chitnis, 2006; Henrique et al., 1997).

The DLL1/RCAS and DLL1/DN tools were already available and have been proven to be efficient to increase NOTCH activity and block NOTCH ligand function, respectively, so we did not repeat the experiments with JAG2.

In any case, since Jag2 is membrane bound, they should show that they have an effect only in cells in direct contact with the Jag2/ Dll1 over-expressing fibers.

The phenotype observed is a decrease in the number of PAX7+ cells upon loss of NOTCH ligand function in differentiated muscle cells (Figure 5). So it will be difficult to visualize a direct contact with the DLL1/DN overexpressing fibres (Tomato) and the absent PAX7+ cells.

*6) It is not clear that Yap activation is downstream of muscle contraction and also that Yap activates Jag2 expression in myofibers. Of course, the most convincing data would be for the authors to show that ectopic activation of YAP in myofibers (via Tol2 MLC electroporation) rescues the loss of PAX7+ progenitors when myofibers have been immobilized. This experiment would considerably strengthen the paper and should be do-able by the authors since they have the Tol2 MLC construct.*

We have now performed the YAP rescue experiments in differentiated muscle cells (using the MLC promoter and the Tol2 stable vectors). We indeed found that forced YAP activity (YapS112A) in differentiated muscle cells (using the MLC promoter) prevented the decrease in the number of PAX7+ cells in immobilization conditions. It also prevented the downregulation of JAG2 transcription in muscles. Moreover, YapS112A activated the expression of ANKRD1 in paralyzed muscles. All these data are now included in a new Figure 7. In agreement with the referees, we believe that this YAP rescue experiment greatly strengthens the conclusion of the paper.

*7) Since YAP is such an ubiquitously expressed molecule, the authors should be careful in the phrasing of the conclusions.*

We have tried to be careful in our conclusions. Although YAP transcripts and protein are observed in many (if not all) cell types, our data nevertheless show that YAP acts downstream of muscle contraction to regulate JAG2 transcription in muscle fibres, which then regulates the pool of muscle progenitors via NOTCH.